# Ozone depletion in the Arctic and Antarctic stratosphere induced by wildfire smoke

Albert Ansmann[1], Kevin Ohneiser[1], Alexandra Chudnovsky[2], Daniel A. Knopf[3], Edwin W. Eloranta[4], Diego Villanueva[5], Patric Seifert[1], Martin Radenz[1], Boris Barja[6], Félix Zamorano[6], Cristofer Jimenez[1], Ronny Engelmann[1], Holger Baars[1], Hannes Griesche[1], Julian Hofer[1], Dietrich Althausen[1], and Ulla Wandinger[1]

[1]Leibniz Institute for Tropospheric Research, Leipzig, Germany
[2]Porter School of the Environment and Earth Sciences, Tel-Aviv University, Tel Aviv, Israel
[3]School of Marine and Atmospheric Sciences, Stony Brook University, Stony Brook, NY 11794-5000, USA
[4]Space Science and Engineering Center, University of Wisconsin, Madison, Wisconsin, USA
[5]Institute for Atmospheric and Climate Science, ETH Zurich, Zurich, Switzerland
[6]Atmospheric Research Laboratory, University of Magallanes, Punta Arenas, Chile

**Correspondence:** A. Ansmann (albert@tropos.de)

**Abstract.** A record-breaking stratospheric ozone loss was observed over the Arctic and Antarctica in 2020. Strong ozone depletion occurred over Antarctica in 2021 as well. The ozone holes developed in smoke-polluted air. In this article, the impact of Siberian and Australian wildfire smoke (dominated by organic aerosol) on the extraordinarily strong ozone reduction is discussed. The study is based on aerosol lidar observations in the North Pole region (October 2019 - May 2020) and over Punta Arenas in southern Chile at 53.2°S (January 2020 - November 2021) as well as on respective NDACC (Network for the Detection of Atmospheric Composition Change) ozone profile observations in the Arctic (Ny-Ålesund) and Antarctica (Neumayer and South Pole stations) in 2020 and 2021. We present a conceptual approach how the smoke may have influenced the formation of polar stratospheric clouds (PSCs) which are of key importance in the ozone-depleting processes. The main results are as follows: (a) The direct impact of wildfire smoke below the PSC height range (at 10-12 km), an additional ozone reduction seems to be similar to the impact of the well-known volcanic sulfate aerosol effects. At heights of 10-12 km, smoke particle surface area (SA) concentrations of 5-7 $\mu m^2\ cm^{-3}$ (Antarctica, spring 2021) and 6-10 $\mu m^2\ cm^{-3}$ (Arctic, spring 2020) were correlated with an ozone reduction in terms of ozone partial pressure of 0.4-1.2 mPa (about 30% further ozone reduction over Antarctica) and of 2-3.5 mPa (Arctic, 20-30% reduction with respect to the long-term springtime mean). (b) Within the PSC height range, we found indications that smoke was able to slightly increase the PSC particle number and surface area concentration. In particular, a smoke-related additional ozone loss of 1-2 mPa (10-20% contribution to the total ozone loss over Antarctica) was observed in the 14-23 km PSC height range in September-October 2020 and 2021. Smoke particle number concentrations ranged from 10 to 100 $cm^{-3}$ and were about a factor of 10 (in 2020) and 5 (in 2021) above the stratospheric aerosol background level. Satellite observations indicated an additional mean column ozone loss (deviation from the long-term mean) of 26-30 Dobson units (9-10%, September 2020, 2021) and 52-57 Dobson units (17-20%, October 2020, 2021) in the smoke-polluted latitudinal Antarctic belt from 70°-80°S.

At heights of 10-12 km, smoke particle surface area (SA) concentrations of 5-7 $\mu$m2 cm-3 (Antarctica, spring 2021) and 6-10 $\mu$m2 cm-3 (Arctic, spring 2020) were correlated with an ozone reduction in terms of ozone partial pressure of 0.4-1.2 mPa (about 30% further ozone reduction over Antarctica) and of 2-3.5 mPa (Arctic, 20-30% reduction with respect to the long-term springtime mean).

## 1   Introduction

Since the summer of 2017, three record-breaking wildfire events occurred, specifically, the Canadian wildfire storm on 12-13 August 2017, the strong, long-lasting Siberian fires in July and August 2019, and the Black Summer fire season of 2019–2020 in southeastern Australia. These events caused major perturbations of the stratospheric aerosol conditions in the Northern Hemisphere (NH) (Khaykin et al., 2018; Peterson et al., 2018; Yu et al., 2019; Hu et al., 2019; Ohneiser et al., 2021) as well as in the Southern Hemisphere (SH) (Ohneiser et al., 2020, 2022a; Khaykin et al., 2020; Kablick et al., 2020). The Canadian wildfire storm led to the largest smoke contamination of the lower stratosphere over Central Europe and the European continent for more than six months in 2017-2018 (Ansmann et al., 2018; Baars et al., 2019; Das et al., 2021). Siberian fires caused an unexpected, dense Arctic smoke layer which was observable over the North Pole region for almost one year (Ohneiser et al., 2021). The Black Summer fire season was finally responsible for the highest smoke-related stratospheric pollution levels ever measured around the globe (Peterson et al., 2021). The January 2020 mean aerosol optical thickness (AOT) for the latitudinal belt from 20°S to 70°S even exceeded its maximum monthly mean AOT observed after the eruption of Mount Pinatubo (Hirsch and Koren, 2021) and significantly influenced the radiation budget in the SH (Hirsch and Koren, 2021; Yu et al., 2021; Stocker et al., 2021; Fasullo et al., 2021; Heinold et al., 2021).

Record-breaking ozone depletion was detected over the Arctic (DeLand et al., 2020; Inness et al., 2020; Wohltmann et al., 2020, 2021; Manney et al., 2020) as well as over Antarctica (Stone et al., 2021; Rieger et al., 2021) in the smoke-polluted stratosphere during the spring seasons of 2020 (March-April in the NH and September-October in the SH). First evidence for an impact of smoke on ozone depletion was found over the High Arctic during the MOSAiC (Multidisciplinary Drifting Observatory for the Study of Arctic Climate) expedition 2019-2020 (Ohneiser et al., 2021; Voosen, 2021). Lidar observations of smoke layering and ozone profiling with sondes were performed aboard the icebreaker Polarstern in the North Pole region during the autumn, winter and spring seasons of 2019-2020 (see Fig. 1 regarding lidar and ozonesonde stations used in this study). A much clearer and unambiguous indication for an influence of smoke on ozone reduction was then obtained in the SH in September-December 2020 by comparing our lidar observations of Australian smoke profiles at Punta Arenas (53.2°S), Chile, at the southernmost tip of South America, with respective stratospheric ozone observations at the NDACC (Network for the Detection of Atmospheric Composition Change) ozonesonde stations at Lauder (45°S), New Zealand, and the two Antarctic Neumayer (70.6°S) and South Pole stations (Ohneiser et al., 2022a). Also Yu et al. (2021), Stone et al. (2021), and Rieger et al. (2021) concluded that the extraordinarily strong ozone loss over the SH polar region in 2020 was probably to a large extent related to the occurrence of Australian smoke in the stratosphere.

The strong ozone hole observed in 2021 was linked to the smoke pollution as well as will be shown in Sects. 5.3 and 5.4. However, Yook et al. (2022) argue that this event in 2021 was caused by sulfate aerosol originating from the La Soufrière volcanic eruption in the Caribbean in the NH in April 2021 (Ravindra Babu et al., 2022), and not by wildfire smoke. Our stratospheric aerosol observations over Punta Arenas in 2020 and 2021 (Ohneiser et al., 2022a) do not support this hypothesis as will be discussed in Sect. 5.3.

It is well known that strong ozone depletion is linked to the development of a cold and long-lasting polar vortex and associated extensive formation of polar stratospheric clouds (PSCs) (Tritscher et al., 2021). PSCs play a key role in the chain of processes leading to the activation of chlorine and bromine components that destroy ozone at sunlight conditions. Most of the conversion of halogen reservoir species into reactive forms takes place on the surface of liquid PSC particles. The impact of background and volcanic sulfate aerosol on PSC, halogen activation, and ozone depletion processes is extensively studied and well known and implemented in climate and ozone forecast models (e.g., Solomon, 1999; Solomon et al., 2015; Zhu et al., 2018). In contrast, the role of wildfire smoke particles in these ozone-depleting processes is unknown and not considered in these models. Our knowledge about the chemical, microphysical, and morphological properties of the aged organic, probably glassy aerosol particles after long range transport around the globe for months or even years is rather poor. Hence, it is unknown how efficient these particles can serve as sites for heterogeneous chemical reactions to produce active halogen components and further chemical processes that lead to a prolongation of the lifetime of the active halogen components (Solomon et al., 2022). It is also unknown how efficiently the smoke particles can act as nuclei in PSC formation processes. In this context, it is noteworthy to add that none of the numerous articles in 2020 and 2021 dealing with the extreme ozone loss over the Arctic in the spring of 2020 mentioned the strongly increased stratospheric aerosol (smoke) burden in their studies and an eventual impact on this record-breaking Arctic ozone hole.

In order to consider smoke-related ozone-depleting processes in existing chemistry climate modeling environments (and thus in the climate change debate), all aspects of this new atmospheric phenomenon need to be explored in detail in upcoming research projects including laboratory studies, airborne in-situ observations, and modeling efforts. The relevance for the required effort is given by the expectation that the number of major fire storms will increase in the twenty-first century due to progressing climate change (Liu et al., 2009, 2014; Abatzoglou et al., 2019; Dowdy et al., 2019). Furthermore, to evaluate the achievements of the Montreal Protocol (Wilka et al., 2021; Feng et al., 2021; Stone et al., 2021), all ozone-loss-relevant influences need to be carefully considered in the analysis on long-term ozone time series. The 1987 Montreal Protocol, and its subsequent amendments during the 1990s, mandated the decrease and eventual cessation of the worldwide production of ozone-depleting substances (ODSs) such as chlorofluorocarbons (CFCs) (Wilka et al., 2021). Within the past few years, ever stronger evidence for global ozone stabilization and a nascent Antarctic ozone recovery has emerged (e.g., Solomon et al., 2016; Weber et al., 2022). Therefore, it is important to identify any deviation from the signatures indicating the long-term healing of the ozone layer and to consider all impacts on ozone depletion in respective modeling studies (Solomon et al., 2022; Bernath et al., 2022).

In this article, for the first time, we systematically investigate the impact of smoke on ozone depletion in the polar stratosphere. We continue and extend the discussion we began in our previous and foregoing publications on the link between aerosol

vertical layering and altitude-dependent ozone losses up to 20-25 km height (Ohneiser et al., 2021, 2022a). The smoke-ozone data analysis is based on the aerosol measurements with two multiwavelength lidars aboard the icebreaker Polarstern and at Punta Arenas and regular ozone profile observations at Ny-Ålesund (78.9°N) in the Arctic and the Neumayer and the South Pole station on the Antarctic continent (see Fig. 1). For an appropriate smoke characterization, we recently developed a wildfire smoke conversion method to derive number and surface area concentrations from the measured stratospheric smoke particle backscatter and extinction profiles (Ansmann et al., 2021a).

The paper is organized as follows. In Sect. 2, we compare the impact of different stratospheric aerosol conditions (background aerosol, volcanic disturbed aerosol conditions, wildfire-smoke-polluted stratosphere) on PSC formation which is of key importance in the ozone-depleting processes. We introduce a conceptual approach how the smoke may influence PSC formation and thus PSC properties. After a brief description of the instruments (lidar, ozonesondes) and data analysis methods in Sect. 3, we continue with the presentation of the main findings regarding the Arctic 2020 and Antarctic 2020 and 2021 ozone depletion seasons in Sects. 4-7. The summary and conclusion remarks are given in Sect. 8.

## 2 PSC formation at background aerosol, volcanic sulfate, and wildfire smoke conditions

Goal of this section is to provide a short updated overview of different stratospheric aerosol scenarios regarding their impact on PSC formation. The information is required to better follow the discussion of findings and observations presented in Sects. 4-7. In the focus is the hypothetical approach how smoke particles may get involved in PSC formation and ozone depleting processes. As shown in the sketch in Fig. 2, we can distinguish three main stratospheric aerosol scenarios: clean background conditions, volcanic events with enhanced sulfate aerosol levels, and situations with high concentrations of wildfire smoke particles (organic particles). The properties of the aerosol and of the PSCs developing in these aerosols are discussed in the following subsections.

### 2.1 Stratospheric background conditions

Tritscher et al. (2021) provides an overview of the aerosol conditions and PSC formation processes in the lower stratosphere at aerosol background conditions. In an undisturbed, clean stratospheric environment, supercooled binary solution (SBS) droplets consisting of water and sulfuric acid predominantly nucleate homogeneously and form the stratospheric background aerosol. Observations show that more than 50% of these sulfate particles within the polar vortex region contain insoluble meteoritic substances (Curtius et al., 2005; Weigel et al., 2014). The particle number concentration is of the order of 5-10 $cm^{-3}$, the median radius of the lognormal size distribution (accumulation mode) close to 100 nm, and the corresponding effective radius about 150 nm (Deshler et al., 2003; English et al., 2011; Zhu et al., 2018).

Strong ozone depletion in polar regions requires the development of a cold, stable, and long-lasting polar vortex with temperatures below 195 K and the formation of extended PSC fields. Upon cooling, the SBS droplets start to grow by uptake of additional $H_2O$ and, when the droplets have become sufficiently dilute, by condensation of $HNO_3$ into the acidic liquid (Carslaw et al., 1994; Koop and Carslaw, 1996; Koop et al., 1997). The median radius of the $H_2O/H_2SO_4/HNO_3$ droplets (also

termed supercooled ternary solution, STS, droplets) increases to around 300 nm (Jumelet et al., 2008) and the effective radius to 400 nm at these aerosol background conditions. The particle number concentration of the liquid PSC particles forming from the SBS particles remain at the low number concentrations of 5-10 cm$^{-3}$. The surface area available for halogen activation, however, increases by a factor of around 10 compared to the one for the SBS size distribution.

Figure 2 illustrates the transition from the background aerosol droplets to the PSC liquid droplets. Our focus is on the liquid PSC particles. Heterogeneous chlorine and bromine activation takes place to about 90% on the surface of the STS droplets (Carslaw et al., 1994; Kawa et al., 1997; Solomon, 1999; Wegner et al., 2012; Kirner et al., 2015), much less on NAT (nitric acid trihydrate) and ice particles. Extensive chemical loss of polar O$_3$ in both hemispheres is, however, always accompanied by an extensive removal of HNO$_3$ (denitrification) by the gravitational settling of NAT particles that are able to grow to large sizes (Solomon, 1999; Fahey et al., 1990, 2001). Dehydration moderates chemical loss of polar ozone as well. Efficient dehydration results from gravitational settling of ice crystals (Tritscher et al., 2021). Orographic influences (e.g., triggering mountain wave evolution) and specific meteorological features are responsible for the year to year varying PSC characteristics and the strength of the springtime ozone depletion and for the differences between ozone reduction over the Arctic and Antarctica.

The stratospheric background particles also initiate NAT particle nucleation and ice formation via heterogeneous nucleation on preexisting foreign nuclei such as the meteoritic material (Curtius et al., 2005; Engel et al., 2013; Hoyle et al., 2013). The sketch in Fig. 2 indicates the inclusion of insoluble meteoritic particle fragments in the SBS and STS droplets by small dots. Heterogeneous ice nucleation on preexisting NAT particles is another pathway of ice formation (Koop et al., 1997b; Voigt et al., 2018).

## 2.2 Volcanic perturbation

It is well documented that volcanic sulfate particles can significantly influence ozone depletion by increasing the particle surface area available for the activation of ozone-destroying halogen components (Hofmann and Solomon, 1989; Hofmann and Oltmans, 1993; Portmann et al., 1996; Ansmann et al., 1996; Solomon, 1999; Solomon et al., 2005, 2016; Dhomse et al., 2015; Stone et al., 2017; Zhu et al., 2018). SO$_2$ plumes injected into the stratosphere after major and moderate volcanic eruptions are converted to H$_2$SO$_4$/H$_2$O aerosol within several weeks to a few months. Volcanic perturbations decline with an e-folding decay time of about 90 days for moderate eruptions (Haywood et al., 2010; Bègue et al., 2017) to 14-16 months for major eruptions (Ansmann et al., 1997; Sekiya et al., 2016).

Figure 2 (center panel) shows how volcanic eruptions can influence the STS droplet population. The increase in aerosol particle and PSC particle number concentration is symbolized by three instead of one particle (stratospheric background scenario). Such an increase in particle number concentration was observed after, e.g., the Sarychev (2009) and Calbuco (2015) eruptions. Enhanced number concentrations ranging from 20-100 cm$^{-3}$, a median radius of about 150 nm, and an effective radius around 200 nm of the accumulation mode particles were found (Mattis et al., 2010; Zhu et al., 2018).

When these numerous volcanic particles are involved in PSC formation, the particles may grow up to about 180-250 nm median radius instead of 300 nm (background case). As a consequence, the overall PSC surface area (liquid particles) available for heterogeneous chemical processes may increase by a factor of 1.2-3.5 for particle number concentrations from 20-100 cm$^{-3}$

compared to the background aerosol scenario (10 particles per cm$^3$). The impact of the particle number concentration on the overall PSC particle surface area was highlighted in detail by Zhu et al. (2018) after the Calbuco volcanic eruption in 2015. Note that the Pinatubo sulfate particle number concentration was clearly below $<$10 cm$^{-3}$ (Deshler et al., 2003) during the winter and spring seasons of 1991/1992 and 1992/1993 so that the background aerosol number concentration remained almost unchanged and thus the PSC properties. According to the simulations of English et al. (2011) the number concentration was even $<$5 cm$^{-3}$ after 6 months after the eruption and of the order of 1-2 cm$^{-3}$ six months later. The Pinatubo volcanic particles were, however, very large with effective radii of 400-600 nm (Ansmann et al., 1997). Note finally that we assume pure volcanic sulfate particles in the sketch in Fig. 2 (center panel) without any insoluble particle fragments (such as fine ash) (Muser et al., 2020; Zhu et al., 2020), which could contribute to the PSC formation process (NAT and ice nucleation) via heterogeneous nucleation pathways.

## 2.3   Wildfire smoke in the stratosphere

Before we focus on the potential impact on the ozone layer, let us briefly summarize how the smoke is transported from the fire sources to the polar regions. Figure 3 provides a schematic overview of the main smoke lofting and transport pathways from the burning areas up to the stratosphere and then within the stratosphere to the high latitudes. Efficient lofting of fresh fire smoke to the upper troposphere and lower stratosphere via strong pyrocumulonimbus (pyroCb) convection (Fromm et al., 2010) occurs within a short time period of 30-120 minutes. Only a small fraction of the smoke particles serve as cloud condensation nuclei and ice-nucleating particles in the cloud towers and a low amount of precipitation is produced in these cloud systems so that only a small fraction of the smoke is scavenged (Rosenfeld et al., 2007). Most of the smoke particles are exhausted through the anvil to the upper troposphere and lower stratosphere (UTLS).

After entering the lower stratosphere, self-lofting processes cause further ascent of the fire smoke layers (Khaykin et al., 2020; Kablick et al., 2020; Torres et al., 2020; Ohneiser et al., 2020). Smoke particles considerably absorb solar radiation and warm the air masses which then ascend. The potential of smoke to ascend over several months is an important aspect that significantly prolongs the residence time of wildfire smoke in the stratosphere (Peterson et al., 2021; Ohneiser et al., 2022a). Khaykin et al. (2020) showed that the Australian smoke ascended from 17 to 35 km within 40 days.

During the long-term travel around the globe the smoke disperses over all mid and high latitudes within a few weeks as reported for Canadian smoke in 2017-2018 (Baars et al., 2019; Das et al., 2021). The Australian smoke reached Antarctica already at the end of January 2020. A coherent smoke cover developed in February-April 2020 (Rieger et al., 2021; Tencé et al., 2022), several months before the polar vortex formed in June. Smoke particles were observed in the height range from the upper troposphere to about 25 km height for about two years (2020-2021) and thus influenced PSC formation typically taking place between 14 and 24 km height.

In the case of Siberian wildfire smoke in July-August 2019 (involved in ozone depletion over the Arctic in 2020), the smoke reached the UTLS region without the assistance by deep cumulonimbus convection. Self-lofting processes in the middle and upper troposphere caused the smoke to ascend towards the tropopause within several days (Ohneiser et al., 2021, 2022b).

### 2.3.1 Chemical composition and aging

Figure 2 (bottom panel) shows how wildfire smoke may disturb the STS droplet evolution in PSCs. In contrast to the impact of sulfate aerosol (background and volcanic conditions), our knowledge about the influence of wildfire smoke on ozone depleting is very poor. As mentioned already, neither the physical and chemical properties of the aged stratospheric smoke particles (after traveling around the globe for several months or even years) nor the potential of these aged smoke particles to influence PSCs evolution and change the PSC properties, and thus halogen activation and ozone depletion are well enough understood. The found coincidence of layers with strongly enhanced smoke pollution levels and significant ozone loss over polar regions (Ohneiser et al., 2021, 2022a) was the motivation to elucidate the potential role of smoke particles in more detail.

Smoke particles from forest fires are largely composed of organic material (organic carbon, OC) and, to a minor part, of black carbon (BC). The BC mass fraction is typically <5% (Dahlkötter et al., 2014; Yu et al., 2019; Torres et al., 2020). Biomass burning aerosol also consists of a complex mixture of organic species including phenolic compounds, organic acids, aromatic molecules, and humic like substances (HULIS) which represent large macromolecules (Lin et al., 2010; Graber and Rudich, 2006; Laskin et al., 2015; Hems et al., 2021). The original structures of the lofted irregularly shaped carbonaceous particles remain widely unchanged when they enter the dry stratosphere via fast pyroCb lofting (Haarig et al., 2018; Ohneiser et al., 2020). Most of the freshly emitted biomass burning particles are fractal-like aggregates consisting of a BC-containing core with an OC coating (China et al., 2013). After months of travel in the stratosphere, during which the particles undergo chemical and physical aging and photo-reaction processes (Hems et al., 2021), the smoke particles seem to be compact and spherical (Baars et al., 2019; Ohneiser et al., 2020). The coated BC particles show most likely a perfect core-shell structure as suggested by Dahlkötter et al. (2014). This shape feature is symbolized in Fig. 2 (wildfire smoke) by the onion-like structure (black BC core, green OC shell). When self-lofting in the troposphere comes into play, aging of smoke (condensation of gases on the smoke particle surfaces) occurs already on the way towards the tropopause. These particles are already compact and spherical in shape when they reach the lower stratosphere (Ohneiser et al., 2021).

At conditions of the UTLS, it can be assumed that the organic species in the biomass burning particles are in a solid (glassy) state (Koop et al., 2011; Shiraiwa et al., 2017; Knopf et al., 2018). The solid phase state is the likely explanation for the long chemical lifetime against multiphase oxidation (Arangio et al., 2015; Knopf et al., 2011; Li et al., 2020; Li and Knopf, 2021). Furthermore, heterogeneous oxidation of glassy organic aerosol particles likely increases their ability to take up water (Slade et al., 2017) and thus contribute to PSC formation and heterogeneous ozone-depleting reactions such as the hydrolysis of $N_2O_5$ (Solomon et al., 2022).

### 2.3.2 Potential smoke impact on PSC formation

The number concentration of stratospheric wildfire smoke particles is in the order of 20-100 cm$^{-3}$ (Ohneiser et al., 2021; Ansmann et al., 2021a), and thus similar to the Calbuco sulfate particle number concentration. The effective radius of the accumulation-mode size distribution of aged smoke particles is around 200 nm, however, sometimes even up to 250-300 nm (Haarig et al., 2018; Ohneiser et al., 2020, 2022a).

Regarding the pathways of the smoke impact on ozone depletion, we hypothesize the following (as illustrated in Fig. 2): The glassy organic smoke particles are able to serve as sink for $H_2O$, $H_2SO_4$, and $HNO_3$ (via condensation of gases on the particles). We further assume that the smoke particles, if coated with a water or a sulfuric-acid solution, are then able to influence PSC formation processes in the same way as volcanic sulfate aerosol. This approach was also selected by Yu et al. (2021) and supported by Solomon et al. (2022) and Yook et al. (2022). Several observations corroborate this assumption. Schill et al. (2020) found an increasing fraction of sulfate on smoke particles with increasing residence time in the middle and upper troposphere. Solomon et al. (2022) concluded from a strong decrease of reactive nitrogen species with increasing stratospheric aerosol extinction, caused by the Australian smoke, over southern midlatitudes that hydrolysis of $N_2O_5$ to form $HNO_3$ took place on the surface of the smoke particles in a manner that is similar to sulfate particles. A prerequisite for these heterogeneous chemical reactions is that the smoke particles are able to take up water on their surfaces. The open question remains to what extent this assumption, that volcanic sulfate particles and the aged stratospheric smoke particles influence PSC formation and heterogeneous chemical reactions in a similar way, holds in reality. This question needs to be clarified in future laboratory work and field studies in combination with modeling efforts. Finally, we cannot exclude that in the glassy or liquid shell of the organic particles further chemical reactions occur that contribute to ozone depletion as well.

For completeness, since the smoke particles are glassy or have at least a solid core part, they may even act as nuclei for NAT and ice crystal formation in PSCs. Under cirrus cloud conditions, solid organic particles can serve as ice-nucleating particles (Murray et al., 2010; Knopf et al., 2018; Engelmann et al., 2021).

## 3 Materials and methods

Figure 1 provides an overview of our lidar stations (Polarstern, Punta Arenas) and the considered NDACC ozonesonde sites (Ny-Ålesund, Neumayer station, South Pole station). Based on the aerosol and ozone profiles measured at these stations, the impact of smoke on ozone depletion is analyzed. Also indicated are the major fire regions in central eastern Siberia and Southeastern Australia, the main sources for the stratospheric aerosol over the High Arctic in the winter half year of 2020 and over the southern mid- to high latitudes in 2020 and 2021. The most intense fires occurred from mid July to mid August 2019 (Siberian fires) and from 28 December 2019 to 5 January 2020 (Australian fires).

### 3.1 Aerosol lidar products

Raman lidar observations in the SH were performed at the campus of the University of Magallanes (UMAG) at Punta Arenas (53.2°S, 70.9°W) from November 2018 to November 2021 in the framework of the DACAPO-PESO (Dynamics, Aerosol, Cloud And Precipitation Observations in the Pristine Environment of the Southern Ocean) campaign (Radenz et al., 2021). The main goal of DACAPO-PESO was the investigation of aerosol–cloud interaction processes in rather pristine and unpolluted tropospheric conditions.

As part of the one-year MOSAiC (September 2019 to October 2020), two advanced lidar instruments, a multiwavelength Raman lidar (Engelmann et al., 2016, 2021; Ohneiser et al., 2021) and a High Spectral Resolution Lidar (HSRL) (Eloranta, 2005),

were operated continuously aboard the drifting German icebreaker Polarstern (Knust, 2017). The ice breaker was trapped in the ice from October 2019 to May 2020 and drifted through the Arctic Ocean at latitudes mainly between 85°N and 88.5°N for seven and a half months (Engelmann et al., 2021). The MOSAiC expedition provided for the first time the unique opportunity to perform lidar observations north of 85°N over the entire winter half year. This part of the Central Arctic is not covered by any regular lidar measurement, neither with ground-based systems nor with the spaceborne CALIOP (Cloud Aerosol Lidar with Orthogonal Polarization) lidar which covers latitudes <81.8°N only.

At Punta Arenas as well as aboard Polarstern, multiwavelength Raman lidars of the Polly type (*POrtabLle Lidar sYstem*) (Engelmann et al., 2016) were used for aerosol profiling (Polly, 2022). The lidar instrument, the measurement channels, and the methods to derive the stratospheric smoke optical properties such as particle backscatter and extinction coefficient, extinction-to-backscatter ratio (lidar ratio), and linear depolarization ratio were presented in previous publications (Ohneiser et al., 2020, 2021, 2022a).

For our study here, we used the 532 nm backscatter coefficients and the measured stratospheric smoke lidar ratios. By multiplying the backscatter coefficients with a characteristic smoke lidar ratio of 85 sr, height profiles of the 532 nm extinction coefficient up to 30 km height were obtained. The extinction coefficients were then converted into particle number and surface-area concentrations by means of a recently introduced conversion scheme for aged stratospheric smoke (Ansmann et al., 2021a). Table 1 shows the lidar products and typical uncertainties in the computed values. The derived particle number concentration $n_{50}$ considers the optically active particles with radius >50 nm. According to Deshler et al. (2003) and Zhu et al. (2018) the total number concentration $n_{\text{tot}}$ of stratospheric particles showing an accumulation mode may be underestimated by a factor of about 1.5-2 when using $n_{50}$ as aerosol proxy for the number concentration.

As discussed in Ansmann et al. (2021a), for aged smoke, a conversion factor of 1.75 Mm $\mu$m$^2$ cm$^{-3}$ was recommended to convert the 532 nm extinction values into surface-area concentrations. This holds for size distributions characterized by effective radii of 220-250 nm. However, we assume that the effective radius shifted towards 200 nm after several months of travel in the stratosphere (as a result of sedimentation and removal of the largest smoke particles). For an effective radius around 200 nm, a conversion factor of 2.5 Mm $\mu$m$^2$ cm$^{-3}$ is more appropriate in the conversion.

In our ozone-related study described in the next subsection, we used winter season (PSC season) mean profiles of the particle number and surface area concentration. To obtain these mean profiles, we averaged the available daily backscatter coefficient profiles (Ohneiser et al., 2021, 2022a), in the first step, before we computed the PSC season mean extinction profile and from this, by applying the conversion parameters, the respective mean surface area and particle number concentration $n_{50}$ of the smoke particles.

In Sect. 6, we show HSRL (lidar) observations to corroborate that the persistent stratospheric aerosol layer over the Arctic was a smoke layer and not a volcanic-sulfate-dominated aerosol layer as suggested in several publications (see, e.g., Kloss et al., 2021a; Gorkavyi et al., 2021). The HSRL is part of the ARM (Atmospheric Radiation Measurement) mobile facility AMF-1 (https://www.arm.gov/capabilities/instruments/hsrl) and was operated side by side with the Polly instrument aboard the Polarstern during the MOSAiC expedition. Similar to the Raman lidar Polly, the HSRL system provides vertical profiles of AOT, particle backscatter coefficient, depolarization ratio, and lidar ratio at 532 nm (Eloranta, 2005; HSRL, 2022).

## 3.2 NDACC ECC ozonesonde profiles

Under the umbrella of NDACC a large number of atmospheric-watch stations are operated that regularly launch ozonesondes to measure the ozone partial pressure up to about 35 km height. Measurements are performed since more than 30 years. All data in recent decades are from electrochemical concentration cell (ECC) ozonesondes, which have a precision of 3%–5% and an overall uncertainty in ozone concentration of about 10% up to 30 km (Wilka et al., 2021). As shown in Fig. 1, we use the ozone profiles of Ny-Ålesund, Norway (78.9°N, 11.9°E), Neumayer station, Antarctica (70.62°S, 8.37°E), and the South Pole station, Antarctica (90°S). Ozone data are collated by the World Ozone and Ultraviolet Data Centre (WOUDC) (WOUDC, 2022).

The strongest ozone reduction caused by activated chlorine and bromine components occurs during the Arctic spring months of March (Mar) and April (Apr) and the Antarctic spring months of September (Sep) and October (Oct). In order to study the impact of wildfire smoke on ozone depletion over Antarctica, we computed the deviation of the September-October mean ozone profile $O_3(z, \text{Sep-Oct}, y)$ for the years $y = 2020$ and 2021 from the respective long-term (2010-2019) mean ozone profile $O_3(z, \text{Sep-Oct}, 2010\text{-}2019)$. The ozone deviation is given by:

$$\Delta O_3(z, \text{Sep-Oct}, y) = O_3(z, \text{Sep-Oct}, y) - O_3(z, \text{Sep-Oct}, 2010\text{-}2019). \tag{1}$$

We used the ozone profiles measured at the Neumayer station and the South Pole station to compute the two-month mean ozone profiles $O_3(z, \text{Sep-Oct}, 2020)$ and $O_3(z, \text{Sep-Oct}, 2021)$ and the long-term mean ozone profile $O_3(z, \text{Sep-Oct}, 2010\text{-}2019)$. The mean (Neumayer + South Pole) ozone profiles represent well the ozone conditions over Antarctica.

We selected a likewise short time period from 2010-2019 in the ozone reference computation to avoid the changing impact of decreasing CFC concentrations (and the corresponding healing of the ozone layer, clearly visible over Antarctica since 2000) on the computed ozone deviations (Solomon et al., 2016; Stone et al., 2021; Rieger et al., 2021; Wilka et al., 2021).

In the same way as for the Antarctic ozone layer, we computed the two-month mean ozone anomaly from the Ny-Ålesund ozonesonde data in the case of the Arctic ozone hole in 2020:

$$\Delta O_3(z, \text{Mar-Apr}, 2020) = O_3(z, \text{Mar-Apr}, 2020) - O_3(z, \text{Mar-Apr}, 2010\text{-}2019). \tag{2}$$

In order to highlight the record-breaking ozone loss in 2020 on the column ozone values and the ozone-layer healing trends, we additionally calculated the deviation of the profile-mean (0-35 km) and monthly mean ozone particle pressure $O_{3,\text{col}}(m, y)$ from the respective long-term monthly mean (2000-2019):

$$\Delta O_{3,\text{col}}(m, y) = O_{3,\text{col}}(m, y) - O_{3,\text{col}}(m, 2000\text{-}2019). \tag{3}$$

Besides the impact of stratospheric aerosol, the temperature plays an important role in the ozone-destroying processes (Rex et al., 2004; Solomon et al., 2015). The potential to convert reservoir into active halogen species is closely related to the available overall PSC volume (Rex et al., 2004) which, in turn, depends on ambient temperatures. To consider temperature effects on the observed ozone loss, we computed the two-month mean deviation of the stratospheric temperature conditions

(and profile structures) during the main PSC seasons (January-February in the Arctic, July-August over Antarctica) for $y = 2020$ and 2021 from the respective long-term means $T(z, \text{Jan-Feb}, 2010\text{-}2019)$ and $T(z, \text{Jul-Aug}, 2010\text{-}2019)$,

$$\Delta T(z, \text{Jul-Aug}, y) = T(z, \text{Jul-Aug}, y) - T(z, \text{Jul-Aug}, 2010\text{-}2019) \tag{4}$$

and

$$\Delta T(z, \text{Jan-Feb}, 2020) = T(z, \text{Jan-Feb}, 2020) - T(z, \text{Jan-Feb}, 2010\text{-}2019). \tag{5}$$

The temperature profiles were measured together with the ozone partial pressures with the NDACC ozonesondes. To highlight the strong variations in the meteorological conditions (influencing the meridional and vertical ozone transport), which have a strong impact on the stratospheric ozone concentration, we show the two-month mean temperature profiles for the individual years from 2010 to 2019 in Sects. 5.4 and 6.3 as well.

We also used ERA5 temperature fields to characterize the meteorological conditions over Antarctica and the Arctic (ERA5, 2022). ERA5 is the fifth generation ECMWF (European Centre for Medium-Range Weather Forecasts) atmospheric reanalysis of the global climate covering the period from January 1950 to present. ERA5 is produced by the Copernicus Climate Change Service at ECMWF. ERA5 provides hourly estimates of a large number of atmospheric, land and oceanic climate variables. We further use column ozone observations from 70°-80°S with the Ozone Monitoring Instrument (OMI) aboard NASA's Aura satellite.

## 4 Arctic and Antarctic column ozone anomalies from 2000-2021: an overview

Figure 4 provides a first glimpse of the record-breaking ozone depletion over Antarctica and the Arctic in 2020. The monthly mean column ozone anomaly $\Delta O_{3,\text{col}}(m, y)$ from the respective long-term monthly mean (2000-2019) as defined in Eq. (3) is shown for the total column up to 35 km height in Figs. 4a and c. The ozonesonde observations at the Neumayer and South Pole stations are used in Fig. 4a and b and Ny-Ålesund observations in Fig. 4c and d. The individual measurements were smoothed with 5-bin (5 consecutive profiles, Antarctica) to 12-bin (12 consecutive profiles, Arctic) temporal window lengths before calculating the monthly mean ozone deviations. Typical remaining variations in the monthly mean column ozone pressure values around the climatological monthly means are ±0.5 mPa over Antarctica and ±1 to ±2 mPa over the Arctic. Ozone depletion is mainly linked to PSC occurrence (between 14 and 23 km height in 2020 and 2021). In addition, we investigated the impact of the strong aerosol perturbation on ozone depletion at widely PSC-free conditions. To that end, we defined the 10-12 km height layer and for this layer the ozone anomalies are shown in Figs. 4b and d. The smoke pollution showed a maximum in the 10-12 km height range over the Arctic and Antarctica in 2020.

The ozone partial pressure varies as a function of meridional and vertical ozone transport and seasonal temperature conditions which determine the PSC volume. Sporadic aerosol events (volcanic eruptions, large wild fires) and associated additional ozone depletion contributes to the variability in the shown ozone deviations. Because of the decreasing CFC levels and the associated recovery of the ozone layer (Solomon et al., 2016; Wilka et al., 2021; Stone et al., 2021; Rieger et al., 2021), expressed in a

long-term increase of the ozone partial pressure, we considered the years from 2010-2019 only (see horizontal line in Figs. 4a and b). Consequently, in the determination of the climatological mean ozone profiles this period was used as reference in the smoke-related ozone depletion study presented in the following sections. Similarly, Rieger et al. (2021) and Yook et al. (2022) considered ozone observations since 2012 only in their smoke-related ozone loss investigations. By averaging of the ozone profiles from 2010-2019, we assume that ozone trends as well as ozone transport effects are widely eliminated and the remaining ozone deviations provide robust hints on the impact of the wildfire smoke on the observed ozone loss in 2020 and 2021.

The most impressive feature in Figs. 4a and b are the two pronounced ozone holes, that developed over Antarctica in a row, i.e., in the springs of 2020 and 2021. These two events coincide with the strong stratospheric perturbation by Australian wildfire smoke in 2020 and 2021 (Black Summer fires, dark gray area in Figs. 4a and b) (Ohneiser et al., 2022a). The ozone holes in 2020 and 2021 belong to the strongest (regarding ozone reduction) and largest (regarding the covered area with very low ozone amount) observed during the last 40 years (Krummel and Fraser, 2021; Stone et al., 2021). The ozone anomalies found for the entire column (up to 35 km) and for the 10-12 km layer are rather similar and suggest that smoke influenced ozone depletion from the lowest part of the stratosphere, where the smoke aerosol concentration was highest, to the upper part of the PSC height range (at 21-23 km height). As we mentioned in the introduction section, Yook et al. (2022) argue that the strong ozone hole over Antarctica in the spring of 2021 was probably caused by the influence of volcanic sulfate aerosol (La Soufrière eruption in the Caribbean in April 2021) and no longer by wildfire smoke. However, our continuous aerosol lidar observations at Punta Arenas observation do not support this hypothesis. A coherent decrease of the smoke perturbation was observed in the lower stratosphere from the beginning of 2020 to the end of 2021 and clearly indicated the dominance of smoke in the southern part of the SH during 2020 and 2021. More details are given in Sect. 5.3.

Two further events of stratospheric aerosol perturbation are indicated by light gray columns in Figs. 4a and b. The impact of the Chilean Calbuco volcanic eruption in April 2015 caused a pronounced ozone anomaly of $< -1$ mPa (from the 2010-2019 monthly mean, shown as horizontal line) in September and October 2015. The Calbuco impact is discussed in detail by Solomon et al. (2016), Zhu et al. (2018), and Rieger et al. (2021). The Australian Black Summer wildfires in 2019-2020 led to similar ozone losses in 2020 and 2021 as the Calbuco aerosol in 2015. In contrast, the Black Saturday fires in February 2009 (Siddaway and Petelina, 2011) had no impact on the ozone layer (see light gray column from Februray 2009 to December 2009 in Fig. 4a and b). Compared to the Australian Black Summer smoke pollution level, the Black Saturday aerosol load was an order of magnitude lower (Peterson et al., 2021).

As shown in Fig. 4c and d, compared to the Antarctic monthly mean ozone anomalies, the respective ozone deviations from the long-term, climatological monthly mean are stronger over the Arctic. Furthermore, a clear trend in the ozone anomalies towards positive values as a result of decreasing CFC levels is not visible in the Arctic ozone data, as already pointed out by Chipperfield et al. (2017). The largest negative monthly mean ozone deviation of about 3 mPa was observed in March 2020 (Fig. 4c). Wohltmann et al. (2020) reported a near-complete local reduction of Arctic stratospheric ozone. The ozonesonde measurements in the most depleted parts of the polar vortex showed ozone losses up to 93-96% at 18 km height mid of March 2020. Inness et al. (2020) confirmed that ozone columns over large parts of the Arctic reached record low values in March

and April 2020. A temporally broad minimum was found in the 10-12 km ozone time series (Fig. 4d). Manney et al. (2020)
emphasized that chlorine activation and ozone depletion began earlier in 2019 than in any previously observed winter and at
lower altitudes, down to 10-12 km. According to DeLand et al. (2020), PSCs occurred over the High Arctic only at heights
>14 km. Thus, remarkable ozone depletion obviously developed even in an PSC-free environment over the Arctic in the winter
half year of 2019-2020.

The Arctic stratosphere was highly polluted with wildfire smoke, mainly from 8 and 18 km height, during the PSC period
(January-April 2020). Major Siberian fires occurring in July-August 2019 were most probably responsible for the aerosol
burden (Ohneiser et al., 2021). Volcanic aerosol originating from the Raikoke volcanic eruption (Kloss et al., 2021a) in June
2019 may have contributed to the stratospheric perturbation by about 10-20% (more details are given in Sect. 6.2). The period
with strongly enhanced stratospheric aerosol pollution levels (August 2019 to May 2020) is indicated by a dark gray column in
Figs. 4b and d. The light gray column (from August 2017 to May 2018) indicates the period with stratospheric smoke from the
Canadian fires in August 2017 (Peterson et al., 2018; Baars et al., 2019). A noticable impact of the Canadian smoke on ozone
depletion is not visible.

In the following sections (Sects. 5-7), we will discuss the additional ozon loss observed during the Australian and Siberian
wildfire smoke periods. We start with the Antarctic ozone holes in 2020 and 2021.

## 5 Antarctic ozone depletion in 2020 and 2021

### 5.1 Spatial column ozone anomalies over Antarctica in September and October 2020 and 2021

In Fig. 5, we show the column ozone anomaly pattern over the southern part of the SH in September and October 2020 and 2021.
We used the Ozone Monitoring Instrument (OMI) on NASA's Aura satellite. The instrument provides daily measurements of
total column ozone with a global daily coverage of most of the Earth's atmosphere. Note that this satellite product includes
both tropospheric and stratospheric ozone. Figure 5 highlights the deviation of the strong ozone hole conditions in 2020 and
2021 from average ozone depletion conditions over Antarctica in September-October 2010-2019. Extraordinarily strong ozone
reduction (blue to black colors) was observed over the entire Antarctic continent in October of both years. Krummel and Fraser
(2021) emphasized the similarities in the 2020 and 2021 ozone hole metrics that were quite striking regarding size, depth,
persistence, and temporal patterns. Fig. 5 corroborates this statement.

Table 2 provides values for the latitudinal (70°-80°S) averaged ozone anomalies in September and October 2020 and 2021.
The column ozone values in October 2020 and 2021 deviated by −50 to −60 Dobson units (DU) from the long-term October
means, or in relative units by 17-20% from the June mean (2010-2019, 70°-80°S) column ozone amount of 280 DU. The
June 2010-2019 ozone values may be regarded as the ozone reservoir available for depletion in September and October. This
additional ozone loss of 17-20% is to a large extent linked to the strong stratospheric aerosol perturbation by the Australian
bushfire smoke.

The results for 2020 in Table 2 are in good agreement with the findings of Rieger et al. (2021). They discussed column
ozone observations in the 13-22 km layer (averaged over the latitudes from 60°-90°S) for the time period from 2012-2020.

Negative ozone anomalies of 20-25 DU from the 2012-2019 October column ozone mean value occurred in October 2020. This corresponds to a relative additional ozone depletion of 14-17% (related to the respective 13-22 km column ozone value for June (2012-2019) of around 145 DU). Yook et al. (2022) extended the aerosol-ozone study for the 60-90°S latitudes to cover 2021 and found similar results in terms of ozone reduction in 2020 and 2021 at slightly lower aerosol extinction coefficients in 2021 in good agreement with our aerosol extinction observations presented in Sect. 5.4.

## 5.2 Height-resolved ozone anomalies (2019-2021) over the Neumayer station

Figure 6 provides a vertically resolved view on ozone depletion in 2020 and 2021. It is highlighted that the two consecutive ozone holes developed in a completely smoke-polluted environment. The ozone profiles of the Neumayer station are used. The deviation of each individual ozone sounding, $O_3(z,t)$ at time $t$, from the long-term monthly mean $O_3(z, m, 2010 - 2019)$ is shown. Base and top heights of the wildfire smoke layer as measured over Punta Arenas (Ohneiser et al., 2022a) are given as gray and black circles. We assume that the smoke was homogeneously distributed over the southern part of the SH in the winter of 2020 (about 6 months after smoke injection) so that the observations at the southernmost tip of South America are representative for the aerosol conditions over the Antarctic continent as well. This assumption is corroborated by aerosol extinction observations for the latitudinal belts from 30°S-60°S and from 60°S-90°S presented by Rieger et al. (2021) and by the extinction observations discussed in Yook et al. (2022). These studies are in good agreement with simulations of the fast spread of volcanic aerosols after mid latitudinal volcanic eruptions (Sarychev, Raikoke, Calbuco) towards polar regions (Haywood et al., 2010; Zhu et al., 2018; Kloss et al., 2021a). Rieger et al. (2021) and Yook et al. (2022) show that the smoke became distributed over the entire southern mid and high latitudes (45-90°S) within the first 2-3 months. The study of Rieger et al. (2021) suggests that the smoke extinction values in the 30°-60°S belt may have been approximately 20% higher than the ones in the 60-90°S zone in June-August 2020.

In Fig. 6, layers from 10 to 12 km (assumed as widely PSC-free zone) and from 14 to 23 km (PSC height range) are marked by horizontal lines. Ozone depletion in these two layers will be analyzed separately in Sect. 5.4. As can be seen, strong negative ozone anomalies (of up to 3-5 mPa partial pressure) are visible between 14 and 23 km height from September-December 2020 and from September-November 2021. Negative ozone deviations (blue colors) dominate in the height range above 14 km from August 2020 to the end of the year 2021. As mentioned, the PSCs formed in smoke-polluted air in both years. Note that the PSC volume over Antarctica is, on average, about a factor of 5 higher than over the Arctic (Pitts et al., 2018; Tritscher et al., 2021).

In the lowest layer (10-12 km), there are a variety of different processes that make it difficult to quantify the smoke-related ozone-depleting effects (Hofmann et al., 1987). It includes horizontal and vertical ozone transport, tropospheric-stratospheric exchange processes, and complex heterogeneous chemical reactions on the surface of the particles. Furthermore, PSCs are frequently observed over Antarctica even at heights down to the tropopause (Pitts et al., 2018; Tritscher et al., 2021).

### 5.3 Contribution of the La Soufriére volcanic aerosol to the Antarctic aerosol burden in 2021

In this section, we will oppose to the Yook et al. (2022) argument that sulfate aerosol originating from the La Soufrière volcanic eruption in the Caribbean in April 2021 (Ravindra Babu et al., 2022) impacted the strong ozone hole over Antarctica in the spring of 2021 rather than wildfire smoke. A volcanic aerosol fraction of the order of 10% in terms of stratospheric AOT is more realistic according to our long-term observations at Punta Arenas (Ohneiser et al., 2022a).

The La Soufrière volcano (13°N, 61°W) erupted on 9 April 2021 and emitted 0.4-0.6 Tg of sulfur dioxide to the lower stratosphere (NASA, 2022). The $SO_2$ mass was converted to 0.6-0.9 Tg sulfate aerosol that caused a hemispheric mean AOT of about 0.005-0.01 according to the well established relationship between emitted $SO_2$ mass, converted sulfate aerosol mass, and related hemispheric AOT (Haywood et al., 2010). Most of the La Soufrière sulfate aerosol remained north of the inner tropical convergence zone (ITCZ) (Ravindra Babu et al., 2022). If we assume that 50% of the $SO_2$ plumes were able to cross the ITCZ and even reach the polar region before the Antarctic vortex formed in 2021, the potential contribution of the volcanic AOT to the overall Antarctic stratospheric AOT was thus 0.0025-0.005. At Punta Arenas, we observed a smoke-related 532 nm AOT of 0.022 in the second half of April 2021 (before La Soufrière sulfate particles could have reached the high southern latitudes). Taking a slow decrease of the wildfire smoke AOT towards 0.02 in July 2021 into account, the volcanic-sulfate-related AOT fraction may have reached values of 10-25% in July 2021. The potential impact of La Soufriére aerosol was hard to identify in our lidar data. We found a slight drop of the backscatter Ångström exponent (532-1064 nm spectral range) from around 2.2-2.3 (before mid May 2021) to around 1.9 (mid May to end of July 2021, not shown as figure here). The hardly visible decrease of the Ångström exponent may be the result of the arrival of fresh volcanic sulfate particles that led to an increase in the large particle fraction of the accumulation mode. The drop in the time series of the backscatter Ångström exponent points to a potential sulfate contribution of about 10-12% to the particle extinction coefficient (and thus to the particle surface area concentration) in July 2021, taking a backscatter Ångström exponent of 1.1 for a fresh volcanic sulfate plume in the stratosphere into account (Mattis et al., 2010).

### 5.4 Aerosol burden and ozone depletion over Antarctica in 2020 and 2021

We began the discussion on a potential impact of wildfire smoke on ozone depletion over Antarctica in the spring of 2020 in the arcticle of Ohneiser et al. (2022a). In Fig. 7, we continue with our data analysis and extend the discussion to the ozone depletion season in 2021. Aerosol, ozone, temperature, and PSC information is presented. Height profiles of the mean particle surface area (SA) concentration and particle number concentration $n_{50}$ (measured at Punta Arenas) for the winter seasons (PSC seasons, June-August 2020 and 2021) are shown together with September-October mean ozone deviations from the long term means, computed by using Eq. (1) and given as red profiles. Neumayer and South Pole ozonesonde data are considered here.

The most striking feature here is the clear correlation between smoke occurrence (and pollution strength) and the extra ozone loss (deviaton from the long term mean), and this in two years in a row. Two strong ozone loss events in consecutive years have never been observed since the Pinatuobo eruption in 1991.

As mentioned in the foregoing section, our lidar observations over Punta Arenas in Figs. 7a and d, are assumed to represent well the stratospheric smoke perturbation even over the high southern latitudes as comparisons with satellite observations indicate (e.g., Kloss et al., 2021b; Rieger et al., 2021; Yook et al., 2022; Sellitto et al., 2022). However, it remains an open question to what extent aerosol concentrations within and outside the polar vortex deviate from each other. Punta Arenas was always outside the polar vortex in 2020 and 2021.

The Arctic MOSAiC lidar observation suggest no big difference between the aerosol burden inside and outside the polar vortex. From October to Decmeber 2019, we found the expected decay of the stratospheric perturbation before the polar vortex formed. From January to March 2019 (within the polar vortex), the aerosol burden was roughly constant. Compared to a scenario with a further steady decrease of the smoke concentration in January-March 2020, the smoke load was higher by about 10-30% in February and March 2020 than expected. The accumulation of particles was obviously caused by the missing horizontal dispersion. This polar vortex effect should be considered in the interpretation of the aerosol profiles shown in Fig. 7a and d. Furthermore, a lowering of the smoke layers by 1-2 km must be considered when using the Punta Arenas aerosol observations to describe the aerosol conditions at 70-90°S (Rieger et al., 2021; Yook et al., 2022).

We also indicate stratospheric background aerosol conditions in Figs. 7a and d based on long-term lidar observations at Lauder, New Zealand, from 1992 to 2015 (Sakai et al., 2016). The background extinction levels were measured during volcanic quiescent times and converted into SA and $n_{50}$ values by using conversion factors for typical background sulfate particle size distributions with an effective radius around 0.15 $\mu$m (Wandinger et al., 1995; Jäger and Deshler, 2002, 2003).

According to Deshler et al. (2003), the $n_{50}$ values (particle number concentrations considering particles with radii >50 nm only) are a factor of 1.5-2 lower than the total particle number concentration (considering all particles, i.e., also particles with radius <50 nm). Clean background conditions are characterized by $n_{50}$ values of 1-5 cm$^{-3}$ and SA concentrations of 0.2-1 $\mu$m$^2$ cm$^{-3}$ in the PSC height range from 14-23 km . These numbers are in good agreement with balloon-borne observations over Laramie (41°N), Wyoming (Deshler et al., 2003) during volcanic quiescent times in the late 1970s (Hofmann and Solomon, 1989) and in the late 1990s (Deshler et al., 2003). In 2020 and 2021, the aerosol SA and $n_{50}$ values were increased by a factor of 5 (2021) to 10 (2020) compared to background conditions. Smoke SA concentrations were of the order of 1-10 $\mu$m$^2$ cm$^{-3}$ (winter 2020) and 1-6 $\mu$m$^2$ cm$^{-3}$ (winter 2021) in the height range from 14-23 km according to the Punta Arenas aerosol observations. The particle number concentration $n_{50}$ increased to 10-60 cm$^{-3}$ in the central PSC height range from 14-23 km in the winter season of 2020 and to 6-25 cm$^{-3}$ in the winter season of 2021. Since the total particle number concentration is about a factor of 1.5-2 higher than the $n_{50}$ values, up to 100 particles per cm$^3$ were available to influence PSC formations and properties in 2020. As discussed in Sect. 2.2, a significant increase in PSC particle number concentrations (STS droplets) can sensitively increase the overall PSC particle surface area concentration Zhu et al. (2018).

Figures 7b and e show the additional ozone loss (red profiles) for September-October 2020 and 2021, i.e., the absolute ozone deviation from the respective long-term September-October (2010-2019) mean profile (see Eq. 1). Neumayer and South Pole station data are used. To highlight the natural variability in the springtime ozone conditions, ozone deviation profiles for the individual years from 2010 to 2019 are given as gray profiles. The variability mainly reflects the varying influence of dynamics (vertical and horizontal transport) and temperatures (PSC volume). The corresponding year-to-year springtime mean

temperature deviations were computed from ERA5 temperature fields for the latitudinal belt from 70-90°S and are shown in Figs. 7c and f (ERA5, 2022). Typical year-to-year springtime ozone variations are of the order of ±1.5 mPa and temperature variations are low (±1 K up to 20 km and±2 K from 20-23 km height). In 2011, an extreme ozone loss was observed at 22.5 km (even larger than the ozone loss in 2020 and 2021). This event is extensively discussed by Solomon et al. (2015).

The impact of the Calbuco volcanic sulfate aerosol on ozone depletion will be used as reference in the discussion of the
520 smoke impact. CALIOP observations in the SH showed that the Calbuco aerosol reached heights up to about 19 km height (Bègue et al., 2017; Zhu et al., 2018) at latitudes >60°S and caused an additional ozone depletion over Antarctica in the height range from 12-19 km (Ivy et al., 2017; Zhu et al., 2018). The Calbuco year 2015 is highlighted by dark gray curves in Figs. 7b, 7c, 7e, and 7f.

The PSC occurrence profile (covering the July-August periods in 2020 and 2021 are added (in Figs. 7c and f) to corroborate
that our separation into 14-23 km (PSC height range) and 10-12km PSC-free height ranges is useful. PSCs typically occur over Antarctica at heights from 12 to 27 km (Pitts et al., 2018; Tritscher et al., 2021). The shown frequency of PSC occurrence was retrieved by using the CALIPSO V4 classification scheme (CALIPSO, 2022; Pitts et al., 2009). All CALIOP data for the Southern Hemisphere during the winter seasons 2020 and 2021 were downloaded and the number of PSC entries obtained with the CALIPSO V4 classification were then computed as a function of height. Below 13 km height cirrus clouds are frequently
misclassified as PSC, therefore the PSC height range is shown down to 13 km only. The PSC frequency distribution did not vary much (±10%) from year to year during the 2015-2021 time period.

### 5.4.1 Discussion

Besides the clear correlation of smoke occurrence and strength in 2020 and 2021 (significantly enhanced values of SA concentration and $n_{50}$, similar layering structures) with the extra ozone loss in the 14-23 km PSC height range in the two consecutive
535 years, we found a pronounced maximum in the additional ozone loss in the uppermost part (around 22.5 km height) of the smoke layer in both years. Here, PSC occurrence was largest. The negative ozone anomalies in both years suggest a significant impact of the presence of smoke on PSC formation processes.

By comparing the smoke-related ozone loss (red curves, Figs. 7b and e) with the Calbuco-sulfate-aerosol-related ozone depletion (dark gray curves, Figs. 7b and e) we observed that the smoke was obviously as efficient as the sulfate particles in
its influence on ozone depletion via the PSC path way in the 16-19 km height range. The sulfate layers produced from the Calbuco $SO_2$ emission were exclusively found below 19-20 km height (Bègue et al., 2017; Zhu et al., 2018). As will be shown in Sect. 7, the Calbuco-related aerosol burden (expressed in SA or $n_{50}$ values) in the 16-19 km layer was equal to the smoke pollution levels observed in 2021. Because ERA5 temperatures were similar in 2015 and 2021 according to Fig. 7f, the extra ozone loss caused by the Calbuco volcanic eruption and the Australian wildfires can be compared. The impact of the Calbuco
volcanic aerosol on ozone depletion is extensively discussed by Solomon et al. (2016), Ivy et al. (2017), and Zhu et al. (2018).

The additional ozone losses ranged from 1-2 mPa (2020) to 0.7-1.7 mPa (2021) in the height range from 14-23 km height and were thus of the order of 5-25% when related to the long-term (2010-2019) mean May-July ozone partial pressure values

(8-15 mPa in the height range from 14-23 km). This additional ozone destruction is in good agreement with our OMI data analysis for the 70-80°S latitudinal belt (Table 2) and the study of Rieger et al. (2021) and Yook et al. (2022).

For the 10-12 km height range, an aerosol-related (extra) ozone loss was not found in the spring of 2020. Ozone transport towards the polar region and ozone loss by chemical heterogeneous reaction on the smoke particles clearly enhanced surface area concentrations (10-12 $\mu m^2$ cm$^{-3}$ in the winter of 2020) obviously compensated each other. In the spring of 2021, enhanced particle SA concentrations of 5-7 $\mu m^2$ cm$^{-3}$ were correlated with an additional ozone loss of 0.4-1.2 mPa. This represents a reduction of the ozone concentration by 30% with respect to the climatological May-July 2010-2019 mean.

As emphasized by Hofmann et al. (1987), ozone loss studies in the 10-12 km layer (close to the tropopause) are generally difficult due to different complex processes that altogether contribute to the natural ozone variations. In particular, stratospheric dynamics, heterogeneous chemical reactions and exchange processes between the ozone-rich stratosphere and ozone-poor troposphere make it challenging to generate such analyses. Even, the impact of PSCs cannot be excluded at heights close to the tropopause as PSCs develop down to these heights over Antarctica. As demonstrated by Hofmann et al. (1987), one can check whether ozone transport effects (introducing abrupt or sudden temporal changes in the ozone mixing ratio and temperature) or heterogeneous chemical processes dominate.

Following the idea of Hofmann et al. (1987), we plotted all individual ozone soundings performed at the Neumayer station from mid August to mid October of both years 2020 and 2021 in Fig. 8. No jumps in the shown ozone mixing ratio were found in both years in the selected layers (10-12 km, 14-16 km, 18-20 km). The smooth decrease in the ozone mixing ratio with time suggests the dominance of heterogeneous chemical processes. The largest decrease of the ozone mixing ratio is given in the PSC height range from 14-16 km, the weakest for the lowest layer (10-12 km) in which the ozone mixing ratio is lowest. In 2021, the influence of chemical processes was a bit stronger in the 10-12 and 18-20 km layers than in 2020, which is probably related to colder ERA5 temperatures (and slightly larger PSC volumes in 2021 than in 2020).

# 6  Arctic ozone depletion in 2020

## 6.1  Height-resolved ozone anomalies (2019-2020) over the Arctic Ny-Ålesund station

In contrast to studies of aerosol effects on polar ozone depletion over Antarctica, investigations of an aerosol-ozone relationship based on observational data over the Arctic regions are rather difficult. Dynamical aspects (meridional and vertical ozone transport) dominate and the strength of the PSC volume and lifetime varies strongly from year to year. In the winter half year 2019-2020 an extremely strong and long-lasting polar vortex developed. According to Wohltmann et al. (2021), the Arctic stratospheric winter 2019/2020 was the coldest winter ever observed in the Arctic stratosphere and showed the lowest ozone mixing ratios ever measured in the Arctic polar vortex. The vortex-averaged ozone loss was one of the largest ever observed in the Arctic. A first discussion of a potential impact of the strong stratospheric aerosol perturbation (dominated by Siberian wildfire smoke) over the Arctic in 2019-2020 on the record-breaking Arctic ozone hole in March-April 2020 was given in Ohneiser et al. (2021) and Voosen (2021).

The potential impact of the Siberian wildfire smoke on the formation of the record breaking ozone depletion in the Arctic in the spring of 2020 is illuminated in Fig. 9. The deviation $\Delta O_3(z,t)$ of each individual ozone sonde observation $O_3(z,t)$ at time $t$ from the long-term monthly mean ozone partial pressure $O_3(z,m,2010-2019)$ is shown. The ozone profiles of the Arctic Ny-Ålesund ozonesonde station (78.9°N), about 800 km south of the Polarstern during the winter and spring months in 2020, are used here. Base and top heights of the main wildfire smoke layer as measured over the Polarstern during the MOSAiC campaign (Ohneiser et al., 2021) are given in this height-time display as gray and black circles. The PSC-free height range (10-12 km) is displayed as well. Only for this height range an aerosol impact on ozone depletion could be studied as outlined below.

As can be seen, negative ozone deviations (blue colors) prevailed since the summer of 2019. Rather strong negative deviations were measured between 16 and 22 km height and later on from 10-16 km height in March and April 2020. The vortex-averaged ozone loss was one of the largest ever observed in the Arctic and are related to record-breaking low temperatures. The strong ozone depletion remained visible (blue colors in the smoke height range) even during the summer of 2020 when meridional ozone transport usually replenished the ozone layer over the Arctic and compensated for the spring time ozone losses. Obviously ozone depletion occurred over large parts of the NH so that such a replenishment was not possible. An impact of ozone-poor tropospheric air on the low ozone values in the 10-12 km height range seems to be negligible. Only in 3 cases out of 27 profile observations the tropopause was above 10 km in the February-April time period. In addition, sudden jumps in the ozone concentrations introduced by sudden changes in the meridional ozone transport were absent in the time series of the ozone mixing ratio in the 10-12 km layer from mid February to mid April indicating the dominance of heterogeneous chemical processes leading to the strong ozone hole.

## 6.2 Smoke identification and contribution of the Raikoke volcanic aerosol to the aerosol burden

As in the case of the Antarctic smoke events and a potential impact of La Soufrière volcanic aerosol on ozone depletion, the Raikoke volcanic impact on Arctic ozone loss needs to be dicussed before we continue our smoke-related ozone study. According to Kloss et al. (2021a), volcanic aerosol originating from the Raikoke volcanic eruption (Kuril Islands, 48°N, 153°E) in June 2019 dominated the stratospheric aerosol burden at mid and high latitudes in the NH in the second half of 2019. However, Ohneiser et al. (2021) showed that these findings did not hold for the High Arctic. Smoke from record-breaking eastern Siberian wildfires fires in July and August 2019 was responsible for most of the aerosol observed over the polar region in the UTLS regime from the late summer of 2019 to the spring of 2020. The Raikoke contribution to the overall particle extinction coefficient at 532 nm was estimated to be of the order of 10-15% (Ohneiser et al., 2021).

Here, we present an alternative approach to estimate the Raikoke fraction. The 550 nm AOT over the Arctic from 60°-90°N reached 0.3 in August 2019, and monthly mean AOTs were 0.13 in September and 0.1 in October 2019. The AOT decreases to 0.07 in the beginning of November 2019 (personal communication, Linlu Wei, University Bremen). These results are retrieved from MODIS observations and by applying a sophisticated analysis scheme (Mei et al., 2020, 2022). CALIOP backscatter profiles indicated a stratospheric 532 nm AOT of 0.1 to 0.15 in the latitudinal belt from 76°-82°N on 19 September 2019. During MOSAiC, we measured stratospheric 532 nm AOTs of 0.08±0.03 in October 2019 at 85°N, 0.06±0.03 in

November 2019, and 0.04±0.02 in January-March 2020 at latitudes from 86-88°N. By assuming a realistic e-folding decay time, describing the exponential decrease of the smoke-related stratospheric perturbation from August to December 2019, of 4 months (Haywood et al., 2010; Ohneiser et al., 2022a), the stratospheric AOT must have been about 0.15-0.2 in the High Arctic in August 2019, in good agreement with the satellite-based observations (personal communication, Linlu Wei) and lidar observations at Ny Ålesund in August 2019 (Ohneiser et al., 2021).

Sulfate aerosol originating from the Raikoke volcanic emission of 1.5-2 Tg $SO_2$ (Gorkavyi et al., 2021; Cai et al., 2022) caused a maximum NH AOT of 0.025-0.03 in August 2019 according to the above mentioned relationship between $SO_2$ mass, sulfate mass, and resulting maximum hemispheric AOT. Thus, the ratio of Raikoke AOT to High Arctic stratospheric AOT was in the range of 0.1-0.2. This estimated AOT fraction of 10-20% is in good agreement with the 10-15% Raikoke contribution as derived from our multiwavelength Polly lidar observations during the MOSAiC campaign (Ohneiser et al., 2021).

Figure 10 shows the Siberian smoke layer observed with the HSRL aboard the Polarstern at 85°-88.5°N in January and February 2020 (HSRL, 2022). The smoke layer is clearly visible between 8 and 16 km height (light blue layer, aerosol traces reached up to 20 km height). PSCs are visible in Fig. 10 at heights above 18 km in January and February 2020. Cirrus virga often developed and were typically found between 2-10 km height. Cirrus formation was triggered by heterogeneous ice nucleation on the smoke particles (Engelmann et al., 2021). Arctic haze (also in dark red) dominated the lidar backscatter signals in the lowest part of the troposphere (Engelmann et al., 2021).

Figure 11 presents a comparison of Polly lidar with HSRL observations (in b, 7-8 February 2020) in terms of particle optical properties. Both lidars were operated side by side aboard the Polarstern vessel during the MOSAiC expedition. The 24 h mean backscatter profile in Figure 11a indicates the pronounced aerosol layer with enhanced backscatter values (above background) up to 20 km height. Polly values of the extinction-to-backscatter ratio (lidar ratio) of around 70 sr (532 nm) and 50 sr (355 nm, not shown here) are clear indications for smoke particles as the main optically active aerosol component (Haarig et al., 2018; Ohneiser et al., 2021; Ansmann et al., 2021b). Volcanic particles typically produce lidar ratios in the 40-55 sr range at both 355 and 532 nm wavelengths (Mattis et al., 2010). The Polly results are corroborated by the HSRL data analysis in Fig. 11b. The height profile of an increasing aerosol optical thickness (AOT, red noisy profile), directly obtained from the measured pure Rayleigh signals, is compared with several AOT profiles obtained by the integration of the measured 532 nm particle backscatter coefficient from 8 km to height $z$ multiplied by a given lidar ratio, which was varied between 20 sr and 80 sr. The best match with the directly observed AOT was found for a typical lidar ratio of 70 sr for aged smoke in the NH.

## 6.3 Aerosol burden and ozone depletion over the Arctic in 2020

In Fig. 12, we present our findings regarding aerosol, ozone, temperature, and PSC conditions over the Arctic in the winter and spring seasons of 2020, in the same way as for the Antarctic ozone depletion events in Fig. 7. Compared to the year-by-year variability in the ozone and temperature deviations (±1.5 mPa, ±1 K in the 10-20 km height range) over Antarctica, the respective ozone and temperature variations are much larger over the Arctic (±3 mPa, ±10 K). This corroborates the general dominance of ozone transport processes in the Arctic.

Because of the record-breaking low temperatures with anomalies of up to $-8.5$ K and the fact that the link between the temperature and PSC volumne is not well enough known, a quantification of the smoke impact on ozone depletion at heights around 18 km is impossible. However, as over Antarctica, the maximum ozone depletion again occurred in the upper part of the smoke layer. According to DeLand et al. (2020) the PSC occurrence maximum moved downward from 21 km in December 2019 to mid-February 2020 to 18 km in March and April 2020 February. Thus, also the Arctic PSCs were to a large extent influenced by the enhanced smoke levels. DeLand et al. (2020) further showed that PSCs occurred above 14 km only so that the defined 10-12 km layer was clearly a PSC-free zone.

Although a remarkably strong correlation between a strongly increased particle surface area of 6-10 $\mu m^2$ $cm^{-3}$ at 10-12 km height and an additional ozone loss of 2-3.5 mPa was observed, a quantification of the smoke contribution to the ozone reduction is not possible. The extraordinarily strong temperature deviations and their unknown impact on ozone depletion (even at PSC free conditions) and the unknown transport of ozone poor air from the 14-20 km layer to the 10-12 km layer are main reasons that prohibit a more detailed study.

The application of the Hofmann approach Hofmann et al. (1987) to the Ny Ålesund observations (not shown here) revealed a temporally constant ozone mixing ratio in the 14-16 km and 10-12 km layers for the main depleting period from mid February to mid April 2020. A decrease of the ozone mixing ratio, indicating the dominating impact of heterogeneous chemical processes, was not visible at heights below 16 km. A reduction was only found for the 18-20 km height range (comparable with the one in Fig. 8 for Antarctica in the spring of 2020). Transport and chemical processes obviously compensated each other in the 10-12 km and 14-16 km layers.

The drop in the ozone partial pressure by 2-3.5 mPa represents an ozone reduction of 20-30% with respect to the 2010-2019 long term mean ozone values of 9 mPa at 10 km to 12.5 mPa at 12 km height. Inness et al. (2020) also reported strong negative ozone anomalies (3-4 mPa) at heights around 10-11 km (200 hPa) over the High Arctic in March-April 2020. Their study was based on Copernicus Atmosphere Monitoring Service (CAMS) reanalysis data (available for the years 2003-2020). As already discussed above, Manney et al. (2020) mentioned that chlorine activation and ozone depletion began earlier in 2019 than in any previous year and occurred at lower altitudes over the High Arctic in 2020 than in other years (2003-2020) except during the cold year of 2011.

## 7 Summarizing discussion: Arctic and Antarctic aerosol layers in 2020 and 2021 and correlation between smoke burden and extra ozone loss

Two summarizing figures are presented in this section. In Fig. 13, the stratospheric smoke conditions over the Arctic in 2020 and over Antarctica in 2020 and 2021 (as estimated from the Punta Arenas observations) are compared with values for the stratospheric aerosol burden over Antarctica in 2015 after the moderate volcanic eruption of the Calbuco volcano (Bègue et al., 2017; Zhu et al., 2018) and over the northern part of the NH in the winter of 1992-1993 after the major Pinatubo eruption (Ansmann et al., 1996, 1997). The impact of volcanic sulfate layers on ozone depletion is meanwhile quite well understood so that the found sulfate-ozone relationships can be used as reference in the characterization of the strength and importance of

680 the smoke aerosol type regarding its ozone-depleting potential. The stratospheric aerosol perturbation by the Calbuco volcanic eruption was comparable with the perturbation by wildfire smoke over Antarctica during the second year (2021). However, the Calbuco volcanic layers reached up to 19-20 km height only.

Very similar smoke layering structures were observed over the Arctic and Antarctica with maximum smoke pollution levels just above the tropopause. The SA concentration and respective $n_{50}$ values (not shown) of the Australian smoke were clearly above the background aerosol level at main PSC heights so that PSC formation was influenced by smoke particles. The PSC
height ranges are given in the left part of Fig. 13. As discussed in the foregoing sections, a maximum in ozone depletion was found in the upper part of the smoke layer and coincided with the PSC volume maximum in all three analysed ozone hole cases (Arctic, 2020, Antarctica 2020 and 2021).

To further illuminate the impact of Australian and Siberian smoke on ozone destruction, correlations between the smoke SA
concentration and the observed additional ozone loss are presented in Fig. 14. The same smoke and ozone data as shown in Figs. 7 and 12 are used. We added two data pairs for volcanic perturbed, PSC-free conditions (big black circles, Calbuco and Pinatubo-related ozone loss values) (Zhu et al., 2018; Ansmann et al., 1996). The Calbuco-related ozone loss was simulated for 70°S in September 2015 and constrained to satellite observations of Calbuco aerosol optical properties (Zhu et al., 2018). In the case of the Pinatubo scenario, we analyzed Ny-Ålesund ozone profile data for March-April 1993 and computed the ozone
deviation from the long-term (March-April, 1998-2008) mean ozone profile and interpreted the found negative ozone anomalies as the contribution of Pinatubo volcanic aerosol to the observed ozone loss in the spring of 1993. The volcanic-induced ozone depletion over Ny-Ålesund was very similar to the ozone depletion values observed over Germany (Lindenberg, near Berlin) (Ansmann et al., 1996).

As can be seen in Fig. 14 (closed triangles), we found an increase of the additional or extra ozone loss with increasing SA
concentration at heights from 10-12 km for the Antarctic observations in 2021 and the Arctic observations in 2020, reasonably in line with the influence of the Calbuco and Pinatubo volcanic aerosol in PSC-free air. For the Antarctic ozone depletion season in 2020, we did not find such a correlation.

For the PSC height range (13-25 km, open triangles), an uncorrelated behavior between the SA concentration or $n_{50}$ for the Antarctic ozone depletion seasons in 2020 as well as in 2021 was obtained. Obviously, it is most important that the stratosphere
is polluted in the PSC formation regime so that the particle number concentration is significantly enhanced compared to background aerosol conditions. Note again, that SA values of 1-10 $\mu m^2$ $cm^{-3}$ indicate total smoke particle number concentrations of 10 to 100 $cm^{-3}$. In both years, the particle number concentration $n_{50}$ was significantly above background level (around a factor of 10 in 2020 and a factor of 5 in 2021).

Finally, the Arctic results (open blue triangles in Fig. 14) remain to be briefly discussed. The strong difference to the
710 Antarctic correlation results (open orange and red triangles) provides a clear impression on the impact of the Arctic temperature anomalies (of up to 8.5 K below) on ozone reduction (of up to 10 mPa). According to the simulation study of Solomon et al. (2015) negative temperature anomalies of up to 8.5 mPa can explain additional ozone losses of up to 8 mPa.

## 8    Summary and conclusions

Two major fire events in Siberia in July-August 2019 and Australia in December 2019 to January 2020 caused record-breaking
stratospheric smoke pollution over both polar regions in 2020 (and over Antarctica in 2021 as well). We presented for the first
time a systematic study of the impact of wildfire smoke on ozone depletion in the polar stratosphere over the Arctic in 2020
and Antarctica in 2020 and 2021. The investigation can be regarded as the continuation of previous investigations started by
Ohneiser et al. (2021, 2022). Our analyses were based on complex height-resolved observations of smoke particle number
and surface area concentrations (over the High Arctic and at the southernmost tip of South America) and Arctic and Antarctic
ozone, temperature and PSC profiling as well as on satellite observations of column ozone in the latitudinal belt from 70°-80°S
in the winter and spring seasons of 2020 and 2021.

Two strong ozone holes (in a row) were observed in 2020 and 2021, in years with significant perturbation of the stratospheric
aerosol conditions by wildfire smoke. Clear indications for a smoke impact on ozone depletion via the PSC pathway was found
over Antarctica in the height range from 14-23 km in both years. The data analysis revealed a smoke-related additional ozone
loss over high southern latitudes of 1-1.5 mPa (10-20%) for smoke particle ensembles characterized by total particle number
concentrations of 10-100 cm$^{-3}$ (factor of about 5-10 above stratospheric aerosol background level). At PSC-free conditions,
the impact of the aged smoke particles was found to be similar to the influence of volcanic sulfate particles. A reduction of
the ozone concentration by 20-30% in the 10-12 km layer, where the maximum smoke pollution layers were observed, was
derived from the ozone profile data.

The extremely unusual atmospheric conditions over the High Arctic in the winter and spring seasons of 2019-2020 with
temperature deviations of up to 8.5 K from the long-term mean made it impossible to properly quantify the impact of wildfire
smoke on PSC formation and subsequent ozone depletion at high northern latitudes.

Satellite observations were used to highlight the ozone depletion on a spatial scale. The satellite data, showing the strong
smoke-related ozone loss over entire Antarctica, were in good agreement with the aerosol and ozone profile studies. The derived
additional column ozone loss (deviation from the long-term mean) ranged from 26-30 Dobson units (9-10%) in September and
52-57 Dobson units in October in the Antarctic latitudinal belt from 70°-80°S in these two years 2020 and 2021 within a
significantly smoke-polluted stratosphere.

Our study may be regarded as a first step and a motivation for further studies on the impact of smoke on ozone depletion.
This will include airborne in situ observations, remote sensing, laboratory studies as well as atmospheric modeling in this
new field of atmospheric science. Many aspects, especially those related to the chemical composition and microphysical and
morphological properties of the aged smoke particles (after traveling around the globe over months and even years) need to
be investigated in detail. The ability of aged smoke particles to influence heterogeneous chemical processes, cirrus and PSC
formation is another important topic of investigations. The relevance for these new studies is given by the expectation that
extraordinarily strong wildfires as a consequence of climate change (higher temperatures, longer droughts) may occur more
often in the future. There is a clear and strong need to accurately consider the impact of wildfire smoke on ozone depletion in

the ozone-layer-healing and future-climate-change debate. The full importance of strong wildfires need to be well considered in upcoming Intergovernmental Panel on Climate Change (IPCC) reports.

## 9  Data availability

Polly lidar observations (level 0 data, measured signals) are in the PollyNet database (Polly, 2022). All the analysis products are available upon request (info@tropos.de). MOSAiC HSRL data is available at http://www.arm.gov/data. Specific analysis products derived from Polarstern HSRL observations can be obtained on request (eloranta@wisc.edu). Basic HSRL overviews are avialble at HSRL (2022). The ozone and temperature data from regular NDACC ozonesonde launches at Ny-Ålesund, Neumayer and South Pole station can be downloaded at the NDACC website (NDACC, 2021) and are available through the World Ozone and Ultraviolet Data Center (WOUDC, 2022). Satellite ozone observations were downloaded from the OMI data base available at OMI (2022). Finally, CALIOP observations of PSCs were used (CALIPSO, 2022).

## 10  Author contributions

The paper was written by AA under strong support by KO, AC, DK, EE, and UW. The data analysis was performed by KO, AC, EE, HB, DV, and RE. PS, MR, CJ, BB, and FZ were involved in the DACAPO-PESO campaign at Punta Arenas and took care of all high quality measurements over the entire 3-year campaign. RE, HG, MR, JH, and DA participated in the MOSAiC field observations aboard Polarstern. All coauthors were actively involved in the extended discussions and the elaboration of the final design of the manuscript.

## 11  Competing interests

The authors declare that they have no conflict of interest.

## 12  Financial support

The authors acknowledge support through the European Research Infrastructure for the observation of Aerosol, Clouds and Trace Gases ACTRIS under grant agreement no. 654109 and 739530 from the European Union's Horizon 2020 research and innovation programme. The field observations at Punta Arenas were partly funded by the German Science Foundation (DFG) project PICNICC with project number 408008112. This research has been supported by the U.S. National Science Foundation (grant no. AGS-1446286) and the U.S. Department of Energy, Office of Science (BER), Atmospheric System Research (grant no. DE-SC0021034). The Polarstern Polly data was produced as part of the international Multidisciplinary drifting Observatory for the Study of the Arctic Climate (MOSAiC) with the tag MOSAiC20192020 and Project ID AWI_PS122_00.

*Acknowledgements.* We are very grateful to Susan Solomon for fruitful and helpful discussion by acting as a reviewer, but also later on via an interactive e-mail discussion. Also thanks to the second reviewer and to Mike Fromm for their critical remarks. The ozonesondes at NDACC stations of Ny-Ålesund and at the Neumayer stations were launched by the Alfred-Wegener-Institut, Helmholtz-Zentrum für Polar-

und Meeresforschung (AWI), Bremerhaven, Germany. NOAA Earth System Research Laboratory, Global Monitoring Division, Boulder, Colorado, U.S.A. is responsible for the South Pole ozonesonde launches. We thank the PIs, Justus Notholt (Ny-Ålesund, Holger Schmithüsen and Peter von der Gathen (Neumayer station), and Bryan Johnson (South Pole station) and all team members involved in carefully performed NDACC ozonesoundings over decades for the unique high-quality ozone data sets. Monthly average ozone values were produced by the NASA Earth Observations team based on data provided by the OMI team. We are grateful to the MOSAiC team and the RV Polarstern

crew for their perfect logistical support during the one-year MOSAiC expedition. We further thank the entire MOSAiC research and logistic teams for their enormous efforts of producing the exemplary and uninterrupted MOSAiC dataset. We are also greatful to the entire research team (University Magellan, Leipzig University, TROPOS) to make the three-year DACAOPO-PESO campaign made the event become a big success. Finally, we are grateful to the CALIPSO team for their well-organized easy-to-use internet platforms.

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

**Table 1.** Overview of Polly observational products and characteristic (typical) relative uncertainties in the determined and retrieved properties. Basic smoke parameters measured with lidar are the particle backscatter coefficient and lidar ratio at 532 nm wavelengths. From these data, extinction coefficients are calculated and converted to surface area and number concentration profiles of the smoke particles. In addition, the ozone partial pressure measured with NDACC ozonesondes and used in our study is listed.

| Parameter | Uncertainty |
|---|---|
| Particle backscatter coefficient $[\text{Mm}^{-1}\ \text{sr}^{-1}]$ | $\leq 10\%$ |
| Particle lidar ratio [sr] | 10-30% |
| Particle surface-area concentration $[\mu\text{m}^2\ \text{cm}^{-3}]$ | 35% |
| Particle number concentration (radius >50 nm) $[\text{cm}^{-3}]$ | 50% |
| Ozone partial pressure [mPa] | 5-10% |

**Table 2.** Monthly mean column ozone anomalies (i.e., deviations from the long-term September and October 2010-2019 mean column ozone values) plus standard deviations for September and October 2020 and 2021. Averaged values for the latitudinal belt from 70°-80°S are given. Relative ozone deviations are related to the respective long-term, monthly mean June (2010-2019, 70°-80°S) column value of 280 Dobson units (DU).

| Month | additional ozone loss | rel. additional ozone loss |
|---|---|---|
| September 2020 | 26±14 DU | 9±5% |
| October 2020 | 52±21 DU | 19±8% |
| September 2021 | 30±18 DU | 11±6% |
| October 2021 | 57±28 DU | 20±10% |

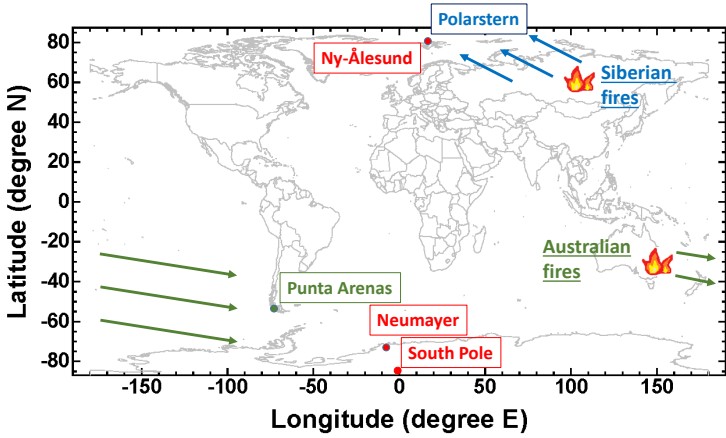

**Figure 1.** Global map showing the two lidar sites (aboard the icebreaker Polarstern during the one-year MOSAiC expedition and at Punta Arenas in southern Chile during the three-year DACAPO-PESO campaign, the regions with major wildfires (Siberia, Australia), and the ozonesonde sites at Ny-Ålesund, Neumayer, and South Pole station. Arrows show the main smoke transport direction towards the Arctic and towards South America and Antarctica during the dispersion phase.

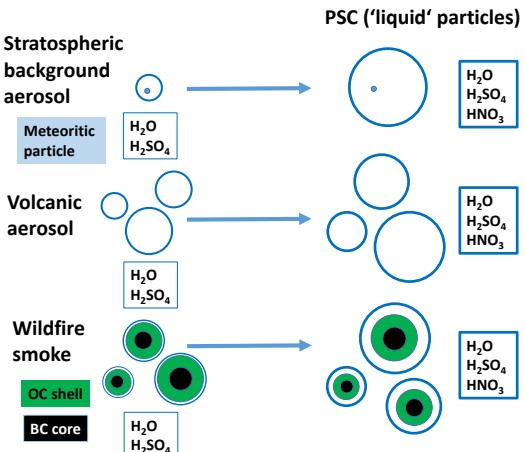

**Figure 2.** Formation of PSC particles (STS droplets) for three different aerosol scenarios (clean background, volcanic aerosol, wildfire smoke). Three aerosol particles symbolize a significant increase in aerosol particle number concentration which leads to an increased STS droplet number concentration and, on average, smaller STS droplets (compared to background conditions). Insoluble meteoritic particles (small dots, background aerosol) may be immersed within the background sulfate and STS droplets. Smoke particles are shown as BC-core-OC-shell structure with $H_2O/H_2SO_4$ coating (white sphere). By $HNO_3$ uptake all STS droplets grow to large sizes (see text for more explanations).

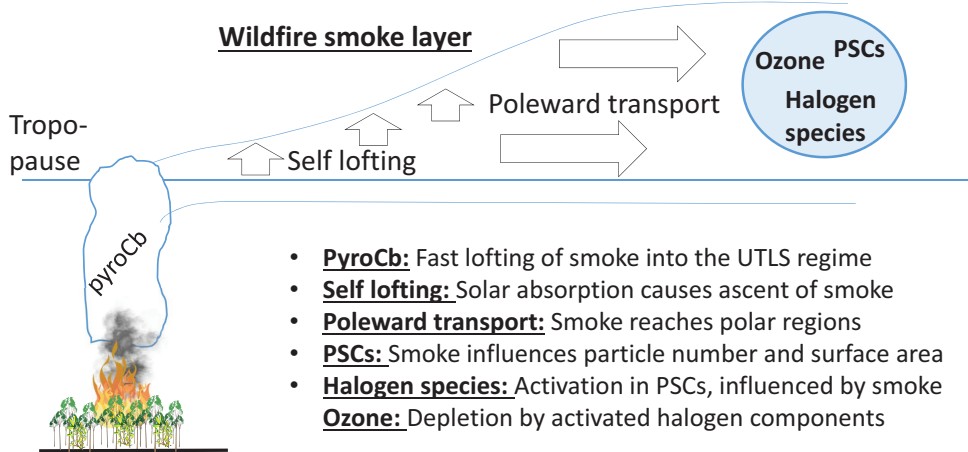

**Wildfire smoke layer**

Tropo-pause

Poleward transport

Self lofting

pyroCb

Ozone PSCs

Halogen species

- **PyroCb:** Fast lofting of smoke into the UTLS regime
- **Self lofting:** Solar absorption causes ascent of smoke
- **Poleward transport:** Smoke reaches polar regions
- **PSCs:** Smoke influences particle number and surface area
- **Halogen species:** Activation in PSCs, influenced by smoke
  **Ozone:** Depletion by activated halogen components

**Figure 3.** Key processes of the vertical and meridional transport of wildfire smoke from the emission sources to the polar regions.

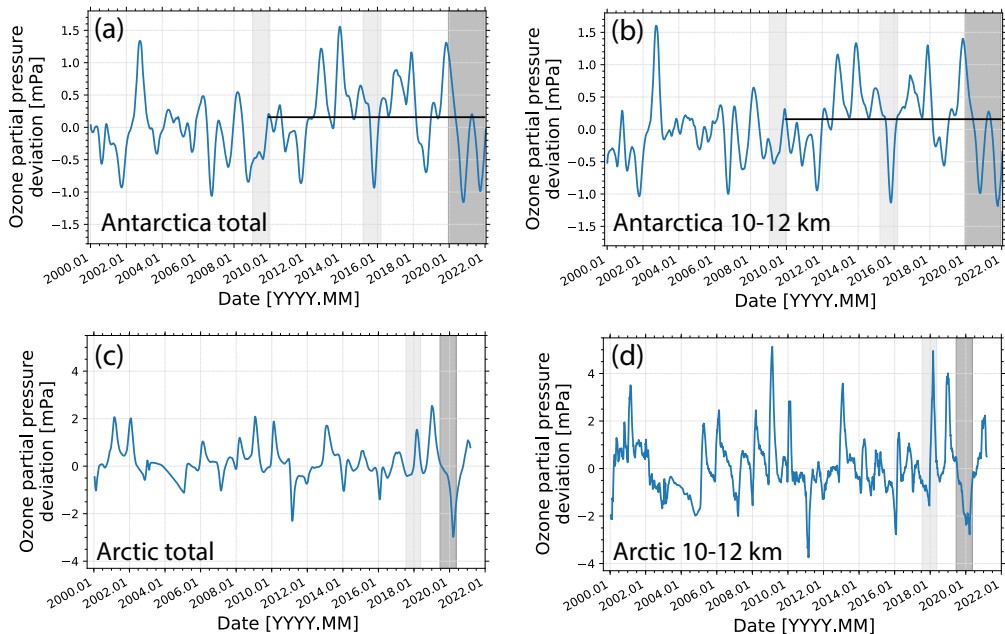

**Figure 4.** (a) Column ozone anomaly expressed as difference between the monthly mean, vertical mean ozone partial pressure (0-35 km height) from the respective climatological (2000-2019) monthly mean (Eq. 3). Ozone profiles measured at the Antarctic Neumayer and South Pole stations are considered, smoothing over 5 observations is applied. (b) same as (a) for the 10-12 km height range. (c) same as (a), except for ozone profiling at the Arctic Ny-Ålesund station. (d) same as (c) for the 10-12 km height range. Dark gray columns mark the time periods with strong stratospheric aerosol perturbations after the major Australian fire events (Black Summer, in a and c) and Siberian fires (in b in d). Light gray columns indicate time periods after the Australian fires in 2009 (Black Saturday, in a and c) and the Calbuco volcanic eruption in April 2015 (in a), and the Canadian fires (in b and d, period of influence from August 2017 to January 2018). The horizontal lines (a, c) emphasizes the change in average ozone level, possibly caused by the healing of the ozone layer during the last years.

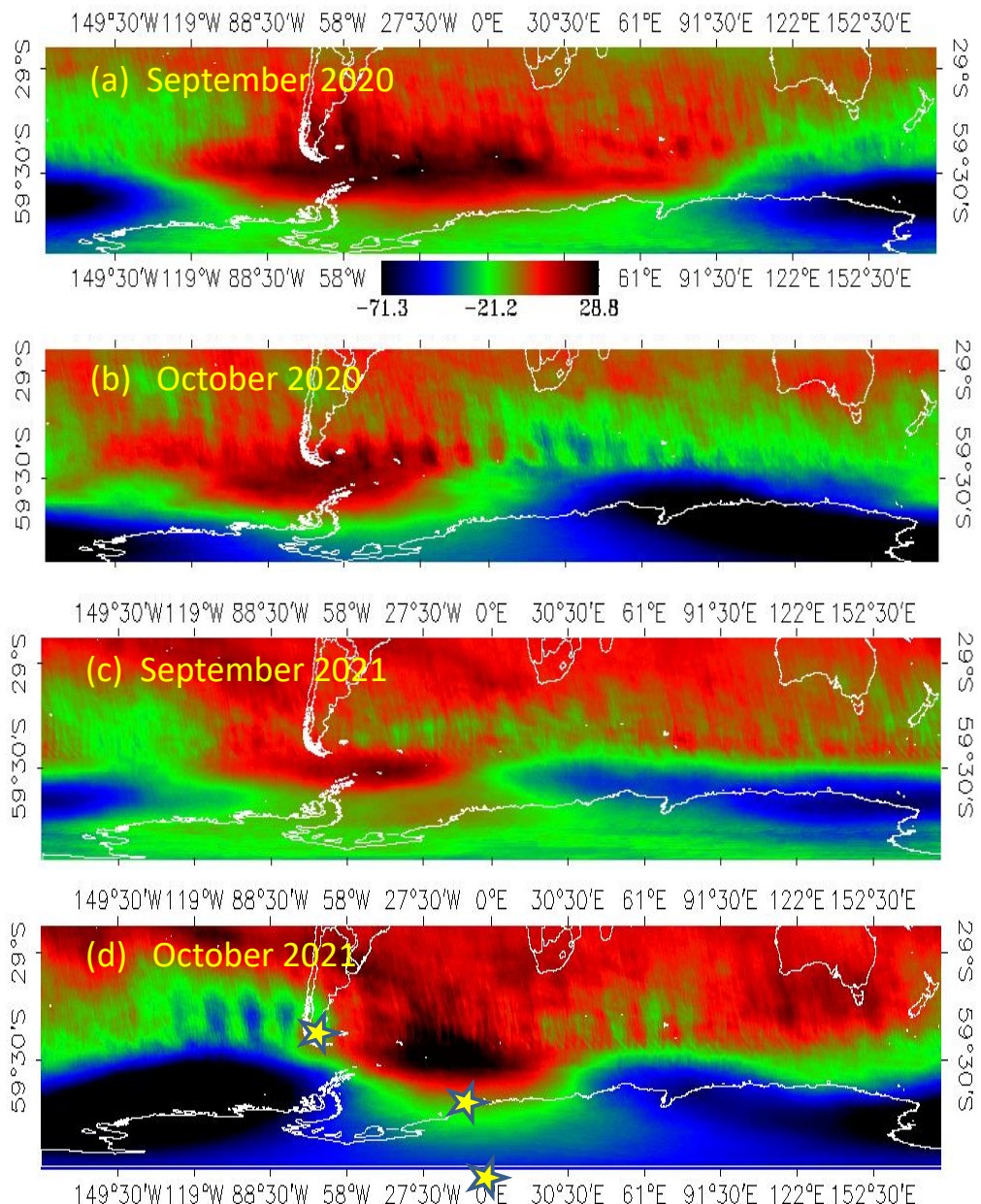

**Figure 5.** (a, c) September and (b, d) October 2020 and 2021 column ozone deviations (monthly means in Dobson units) from the respective September and October 2010-2019 column ozone means. Observations of the Ozone Monitoring Instrument (OMI) aboard the NASA Aura spacecraft are used (OMI, 2022). Both years are similar regarding the additional ozone loss pattern over Antarctica (green, blue and black colors). In panel d, Punta Arenas (53.2°S), the Neumayer (70.6°N) and the South Pole stations are indicated by yellow stars.

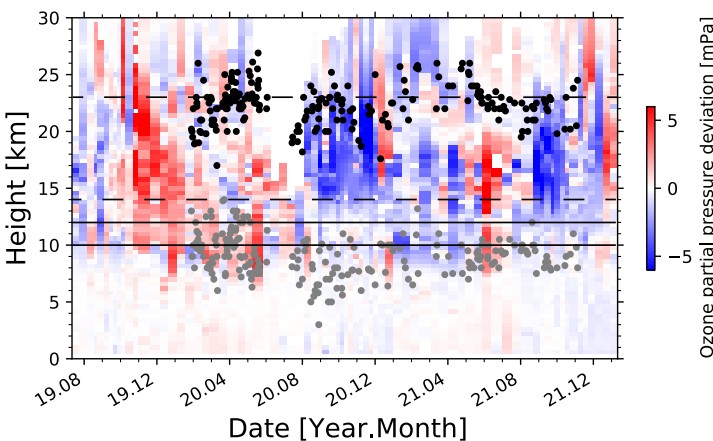

**Figure 6.** Deviation of each individual ozone profile from the respective long-term (2010-2019) monthly mean ozone profile. Measurements at the Neumayer station (70.6°S) are used. Dashed and solid horizontal lines mark the main PSC height range (14-23 km) and the PSC-free zone (10-12 km). The base (gray dots) and top heights (black dots) of the Australian smoke layer measured with Polly at Punta Arenas on a daily basis indicate the smoke-polluted height range. Gaps in the lidar data after December 2019 are caused by cloudy weather and instrumental problems.

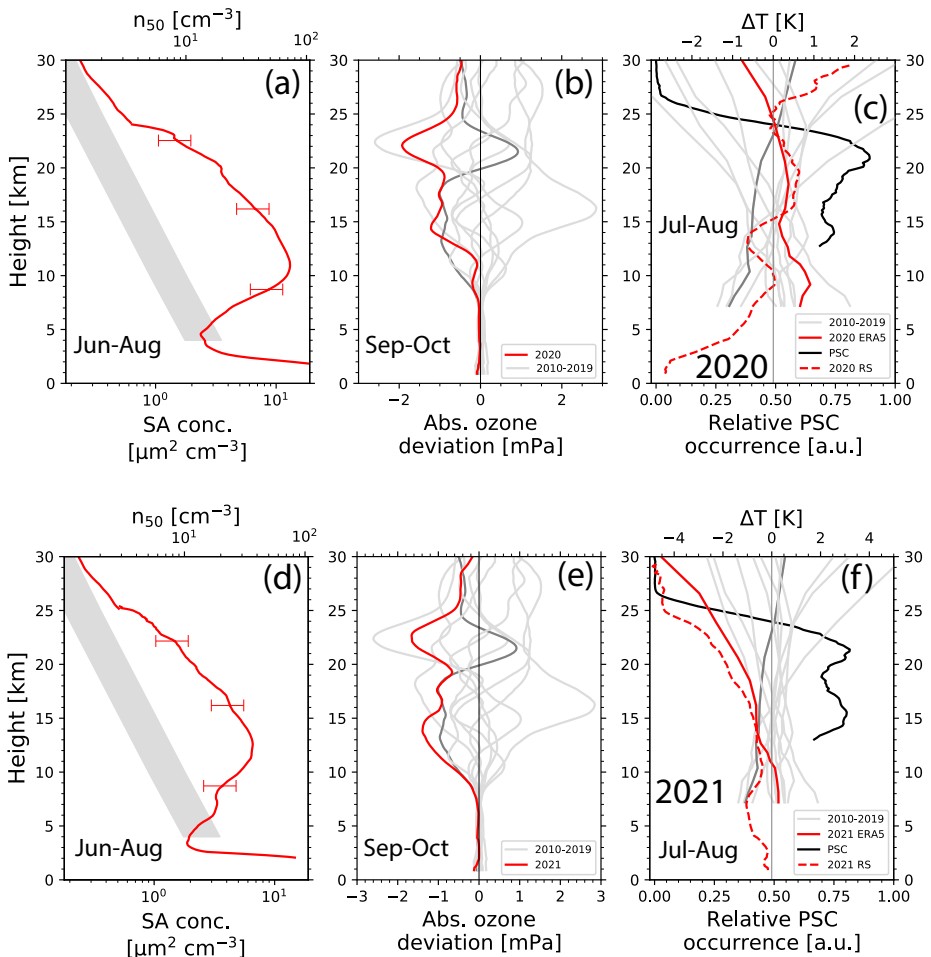

**Figure 7.** (a) Winter (June-August) mean profiles of the particle surface area (SA) concentration and number concentration $n_{50}$ (particles with radius >50 nm) estimated from lidar observations at Punta Arenas in 2020. Horizontal red bars indicate a SA retrieval uncertainty of 35%. Background aerosol conditions are given as a gray shaded area (based on lidar observations at Lauder, New Zealand) (Sakai et al., 2016). (b) Additional ozone loss (red profile) for September-October 2020, i.e., absolute ozone deviations from the respective long-term September-October (2010-2019) mean profile (see Eq. 1). Respective ozone deviations for the individual years from 2010-2019 are given as light gray profiles, the 2015 deviations (Calbuco year) are highlighted as dark gray profile. (c) PSC height range (restricted to heights >13 km) and relative vertical distribution of the PSC frequency of occurrence (in arbitrary units) from CALIOP observations and mean temperature deviations (2020 ERA5 mean values for the 70-90°S latitudinal belt as dashed red profile (ERA5, 2022), ozonesonde/radiosonde (2020 RS) data as red solid profile) for the main PSC months July-August 2020 from the respective July-August long-term mean profile (calculated with Eq. 4). Again, respective temperature deviation profiles (based on sonde data) for the individual years from 2010-2019 are given as light gray lines, the 2015 profile (Calbuco eruption) is highlighted as dark gray line. Ozondesonde data collected at the Neumayer and South Pole stations are used (averaged) in the ozone and temperature calculations (2020 RS data). (d)-(f) same as (a)-(c) except for 2021.

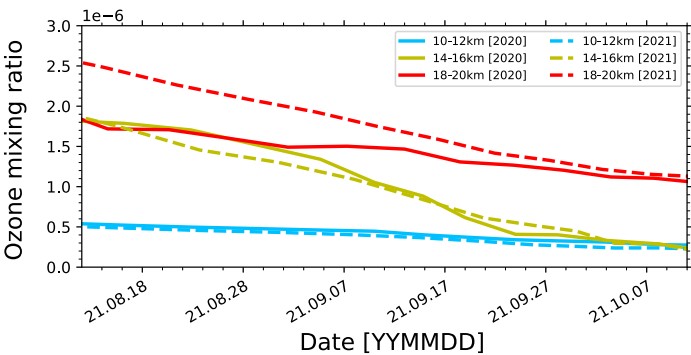

**Figure 8.** Temporal development of the ozone mixing ratio. Observations at the Neumayer site from mid August to mid October in 2020 (dashed line) and 2021 (solid line) are used.

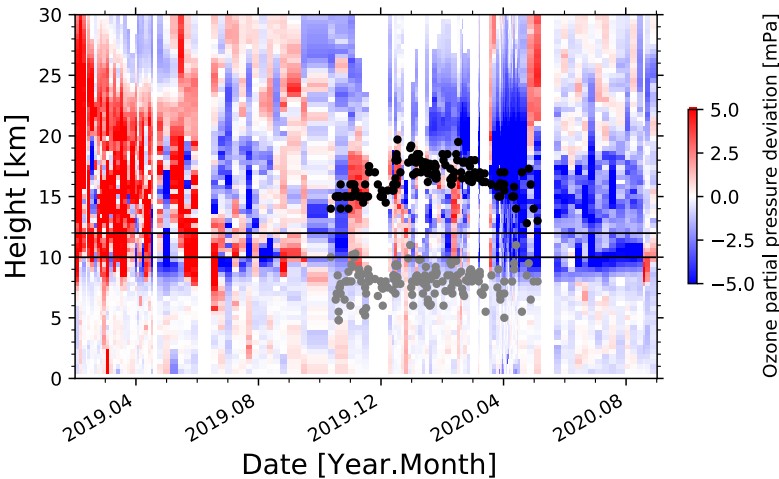

**Figure 9.** Deviation of each individual ozone profile from the respective long-term (2010-2019) monthly mean ozone profile. Measurements at the Ny-Ålesund (78.9°N) station are used. The solid horizontal lines mark the PSC-free height range (10-12 km). The base (gray dots) and top heights (black dots) of the Siberian smoke layer measured with Polly aboard Polarstern on a daily basis indicate the smoke-polluted height range.

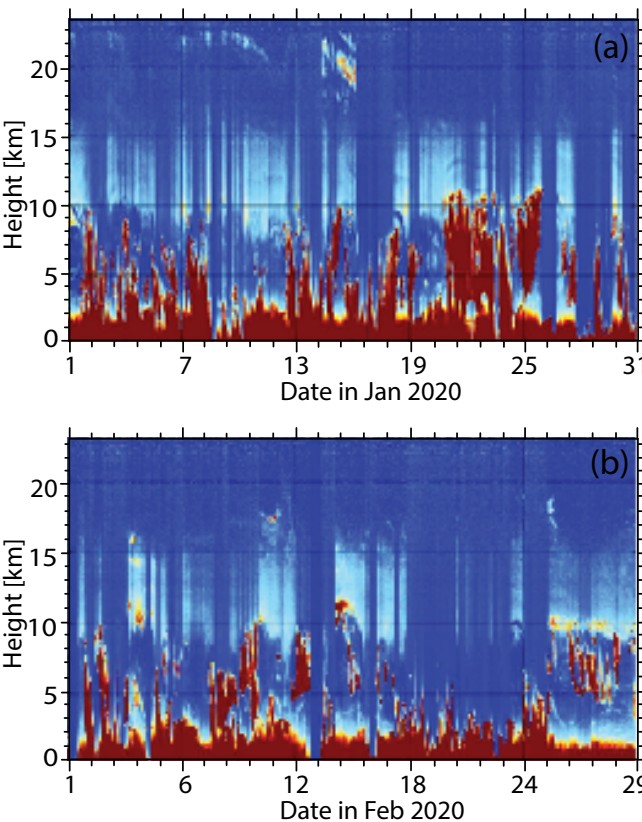

**Figure 10.** UTLS aerosol layer (light blue layer, 8-16 km) over the North Pole region in (a) January 2020 and (b) February 2020. A few PSCs are visible above 18 km in January and between 15 and 20 km in February. Cirrus (in red) formed in the lower part of the UTLS smoke layer and produced long virga (in red). Below 3 km, Arctic haze prevailed. The range-corrected 532 nm backscatter signal observed with a High Spectral Resolution Lidar aboard Polarstern is shown (HSRL, 2022).

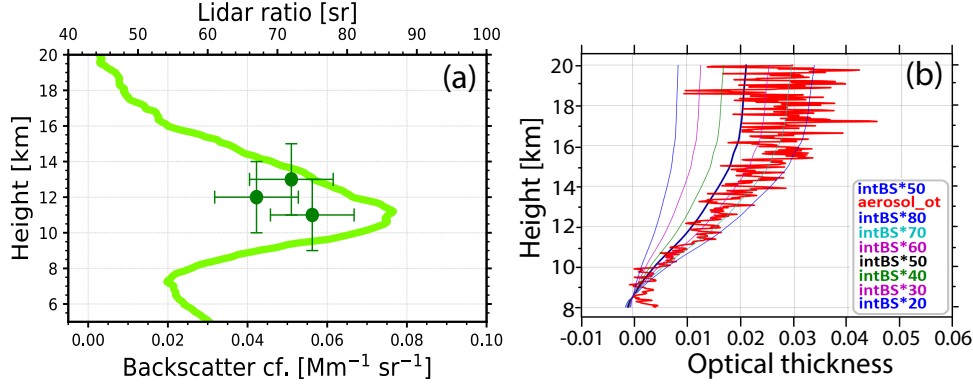

**Figure 11.** (a) Particle backscatter coefficient (light green, 532 nm, 500 m vertical signal smoothing) and corresponding lidar ratio (dark green circles, 4000 m vertical smoothing, smoothing range is indicated by vertical bars) determined from the Polly observations on 7 February 2020, 00:00-24:00 UTC. Horizontal bars indicate one standard deviation uncertainty. (b) Profile of particle optical thickness (thick red profile) from 8 km up to height $z$ as directly measured with the HSRL at 532 nm (aerosol_ot) and, alternatively calculated from the integral (intBS) of the independently measured backscatter coefficients (above 8 km height up to height $z$) multiplied by seven different lidar ratios from 20-80 sr. The best match is obtained for a typical wildfire smoke lidar ratio of 70 sr. HSRL signal profiles are averaged for the time period from 7 February 00:00 UTC to 8 February, 05:50 UTC.

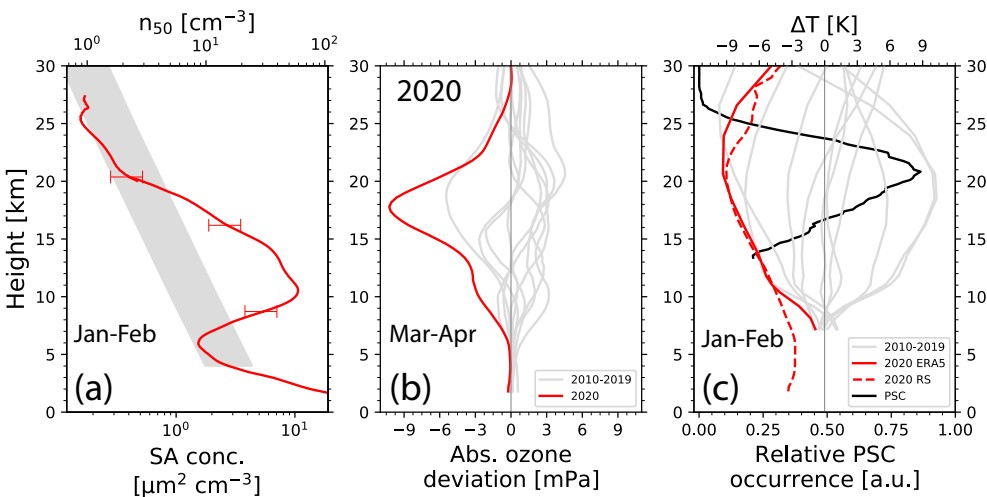

**Figure 12.** (a) Winter (January-February) mean profiles of the particle surface area (SA) concentration and number concentration $n_{50}$ (particles with radius >50 nm) estimated from lidar observations aboard Polarstern in 2020. Horizontal red bars indicate a SA retrieval uncertainty of 35%. Background aerosol conditions are given as a gray shaded area (based on lidar observations at Tsukuba, Japan) (Sakai et al., 2016). (b) Additional ozone loss (red profile) for March-April 2020, i.e., absolute ozone deviations from the respective long-term March-April (2010-2019) mean ozone profile (see Eq. 2). Respective ozone deviations for the individual years from 2010-2019 are given as light gray profiles. (c) PSC height range (restricted to heights >13 km) and relative vertical distribution of the PSC frequency of occurrence (in arbitrary units) from CALIOP observations and mean temperature deviations (2020 ERA5 mean values for the 70-90°N latitudinal belt as dashed red profile, ozonesonde/radiosonde (2020 RS) data as red solid profile) for the main PSC months January-Februray 2020 from the respective January-February long-term mean profile (calculated with Eq. 5). Ozondesonde data collected at Ny-Ålesund are used in the ozone and temperature calculations (2020 RS data).

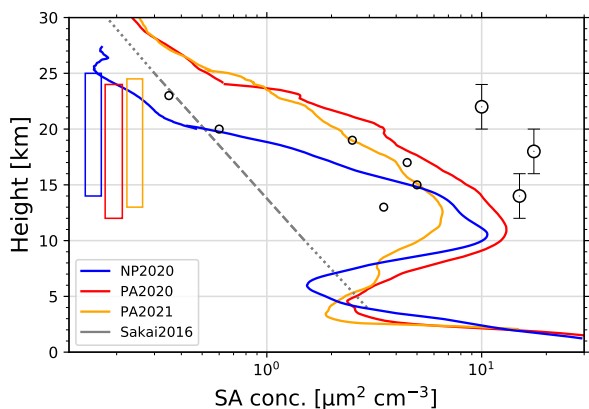

**Figure 13.** Arctic (NP2020, North Pole, January-February 2020) and Antarctic (PA2020, PA2021, Punta Arenas, June-August 2020, 2021) wildfire smoke layers in terms of particle surface area (SA) concentration. The background aerosol SA concentration is given as well (Sakai et al., 2016). For comparison, SA concentrations observed a few months after the minor Calbuco volcanic eruption (small circles, derived from satellite observations at 55°S, July 2015) (Zhu et al., 2018) and about 1.5 years after the major Pinatubo volcanic eruption (big circles, mean SA concentrations for the height range indicated by vertical bars, 53.4°N, Germany, spring 1993) (Ansmann et al., 1996) are included. PSC occurrence height ranges (open vertical bars to the left) for the Arctic (NP2020, blue) and Antarctic winter seasons (PA2020, red, PA2021, orange) are given as well.

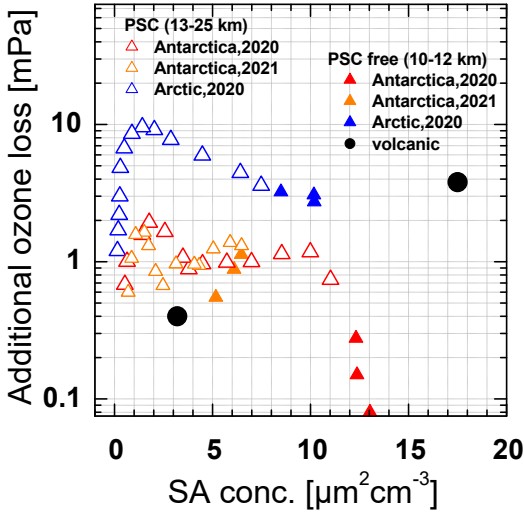

**Figure 14.** Correlation between smoke particle surface area (SA) concentration and the additional ozone loss observed over the Arctic (blue) and over Antarctica (red, orange). The smoke-ozone data pairs are taken from Figs. 7 and 12. Considered are data from 10-12 km (PSC free height range) and from 13-25 km height (PSC height range) with a height resolution of 1 km. For comparison, two values (for PSC free conditions) for volcanic aerosol scenarios are included (Calbuco, low SA conc., Zhu et al. (2018), Pinatubo, high SA conc., Ansmann et al. (1996)).