# Peer review of "Ozone depletion in the Arctic and Antarctic stratosphere induced by wildfire smoke"

_Atmospheric Chemistry and Physics, 2022_

## Referee Comment (RC1)

General comments: I enjoyed reading this paper and think it would make a useful contribution to the literature.

My comments in order of occurrence are:

1) I do think the paper could be made substantially shorter, with a tighter focus on the data itself and what is new (as opposed to review – while I would love to see a long review of wildfires by these authors published in another form, I think including so much background here is distracting and weakens the utility of their new material). In addition, the way wildfire smoke interacts with PSCs or volcanic sulfate is unknown and the speculation here may be right but we really don't know. I think the material on pages 1-8, and Figures 2 and 3, would be part of a great review paper.

2) Line 294. Please specify what dataset(s) was used for this averaging; several were mentioned.

3) Figure 4 is useful, but since a key focus of the paper is the changes at altitudes too low for normal PSC chemistry, I think it would be very valuable to add two more panels showing the same thing for the ozone in the 9-12 km region that is highlighted in the abstract and elsewhere. We really need to see how unusual the reported changes in that height range are, compared to other years. This should be presented either here or elsewhere. Also, can 2021 be added to this plot?

4) Line 364. Near-complete reduction at what vertical level?

5) Line 371. Reference is needed for the statement that Raikoke contributed 10%, or say "(see below)".

6) Can Figure 6 be hatched to show values that are the lowest in the record (if there are any)? Also, how can we relate this figure to statements made elsewhere regarding the 9-12 km layer?

7) It would be great if Figure 7 could delineate the degree to which the high aerosol amounts in the lower portion of the profile are unusual. Would it be possible to show the range of previous years as a shaded region rather than just the single line for the mean?

8) Again re Figure 7, how about showing the range of temperature anomaly in 2020 and 2021 from reanalysis? Single station data can sometimes be unrepresentative.

9) Same issue for the low altitude ozone anomaly in Figure 7 – how unusual is it? A time series like what I suggested in Figure 4 would best answer that question in my view; alternatively, a range of variability in other years could be shown here.

10) Lines 460-463. This is a very important point – can it be expanded and substantiated?

11) The discussion of how much extra ozone loss occurs in the PSC region (lines 479-484) is pretty rough. Factors such as temperature and transport can induce

variability in this region from year to year.   This discussion is not very convincing as a result.   Can you put this figure in context relative to variability from other factors?  If not, I suggest deleting it.

12) Line 505.  Please clarify how you identify the PSCs only about 18 km.   How can you be sure?

13) Line 521.   I see your broad points here but I don't follow how you get the fraction volcanic – yes, the ratio would likely have been about 45 for volcanic but was measured to be 70.   But please explain how you get 10%.

14) Figure 10.   Same comment as above re. the corresponding Antarctic plot, Fig 4.

15) Line 543.   Same problem with variability as made above re the Antarctic; I don't think you can reliably infer the extra ozone loss in this way.   For example, cold years also imply very little downwelling, which will also lead to low ozone values, and I think you can't be confident from such a simple way of looking at it.

16) The low altitude anomalies are key in Figure 11.   Same comments as above for the Antarctic – we need to better understand how unusual these are.

17) In closing, I want to make one more major comment.   There is evidence from OMPS data and presentation on that which I have seen that the aerosols from the Soufriere eruption in 2021 may have contributed to the aerosol load in the Antarctic in that year.    It would on the other hand, be surprising if so much of the 2020 Australian fire aerosol lasted into 2021.    You may want to make the same kind of comparison that you did for the Arctic in Figure 9 to probe the extinction and what that can tell you.   Or you may want to at least discuss the possibility.

---

## Community Comment (CC1)

Ansmann et al. (2022) (hereafter A22) focus equally on Antarctic and Arctic ozone depletion and its proffered connection to stratospheric wildfire smoke. This comment offers questions/concerns with both aspects.

Regarding the Antarctic aspect of A22, the only smoke-particle information presented is from Punta Arenas. At 53°S, this location is generally on the extratropical side of the Antarctic polar vortex. Smoke observations at such a position with respect to the polar vortex interior provide no information on aerosol conditions inside the vortex edge, an effective mixing barrier according to volumes of previous research. To make a plausible link between Pinta Arenas stratospheric smoke and PSCs, the authors may need to demonstrate evidence of smoke inside the vortex, especially in the days leading up to PSC formation (typically late May). Any observation at Punta Arenas that is outside the vortex edge has no direct bearing on PSCs and PSC-related ozone depletion. It would be very interesting and important to see Punta Arenas lidar profiles segregated by their location with respect to the vortex edge. If inside-vortex aerosol enhancements consistent with the pyroCb smoke signature can be demonstrated, in air too warm for PSC support, that would immeasurably advance A22's argument.

Regarding the Arctic aspect of A22, they build on conclusions drawn by Ohneiser et al. (2021) (O21), namely that smoke polluted the Arctic stratosphere in 2019 and 2020. O21 hypothesized that smoke arrived in the stratosphere by a non-pyroCb pathway; Siberian smoke was diabatically lofted from the lower troposphere across the tropopause and continued its diabatic ascent thereafter. O21's confident determination of smoke altitude extent, based on the Polly lidar-ratio calculation, was capped at 12-13 km. A22 loosen that constrain in section 4.3.1: "Before we can deepen this discussion, we need to clarify that wildfire smoke was the dominating aerosol component throughout the entire stratospheric aerosol layer up to 18 or even 20 km height. Ohneiser et al. (2021) left the question open whether smoke or sulfate aerosol originating from the Raikoke volcanic eruption (Kloss et al., 2021) was prevailing at heights >13 km. Because of too noisy Polly lidar signals, wildfire smoke could unambiguously be identified up to 13 km height only." A22's presentation of Figures 8 an 9 appears to be an argument toward a reinterpretation of O21's smoke-altitude cap. The argument is based largely on an admittedly noisy HSRL AOD profile and a set of generated AOD curves spanning a selection of lidar ratios. Figure 9b reveals that a lidar-ratio selection of 60 would give a solution with significant HSRL-data overlap throughout the displayed z range. Prata et al., (2017) demonstrated a 532 nm lidar-ratio central tendency between 59-66 sr in an analysis of two distinct stratospheric volcanic sulfate plumes (Kasatochi and Sarychev Peak). Considering the lidar-ratio error bars in Figure 9a, Prata et al. (2017), and Mattis et al. (2010), it appears that sulfates could explain the HSRL AOD profile in Figure 9b as convincingly as smoke. Given the undisputed evidence that Raikoke sulfates were a hemispheric, high northern latitude presence in 2019 and 2020 (Kloss et al., (2021);

Gorkavyi et al. (2021); Cameron et al. (2021)), shouldn't equal weight be given to their establishment in the Arctic during MOSAiC?

References:

Cameron, W., Bernath, P., and Boone, C.: Sulfur dioxide from the atmospheric chemistry experiment (ACE) satellite, J. Quant. Spectrosc. Ra., 258, 107341, https://doi.org/10.1016/j.jqsrt.2020.107341, 2021.

Gorkavyi, N., Krotkov, N., Li, C., Lait, L., Colarco, P., Carn, S., DeLand, M., Newman, P., Schoeberl, M., Taha, G., Torres, O., Vasilkov, A., and Joiner, J.: Tracking aerosols and $SO_2$ clouds from the Raikoke eruption: 3D view from satellite observations, Atmos. Meas. Tech., 14, 7545–7563, https://doi.org/10.5194/amt-14-7545-2021, 2021.

Kloss, C., Berthet, G., Sellitto, P., Ploeger, F., Taha, G., Tidiga, M., Eremenko, M., Bossolasco, A., Jégou, F., Renard, J.-B., and Legras, B.: Stratospheric aerosol layer perturbation caused by the 2019 Raikoke and Ulawun eruptions and their radiative forcing, Atmos. Chem. Phys., 21, 535–560, https://doi.org/10.5194/acp-21-535-2021, 2021.

Prata, A. T., Young, S. A., Siems, S. T., and Manton, M. J.: Lidar ratios of stratospheric volcanic ash and sulfate aerosols retrieved from CALIOP measurements, Atmos. Chem. Phys., 17, 8599–8618, https://doi.org/10.5194/acp-17-8599-2017, 2017.

---

## Author Comment (AC1)

Dear Prof. Solomon,

We sincerely thank you for a very careful reading, all the critical comments and fruitful suggestions from the point of an absolute expert in the field.

We rearranged the results section (more sections, no longer sub subsections) and in the revised version considered all your comments.

Our answers are in BLUE!

Note that manuscript text that was substantially changed or added, appears in BOLD in the revised version.

General comments:   I enjoyed reading this paper and think it would make a useful contribution to the literature.

We are pleased to know that you like our manuscript. Thank you!

My comments in order of occurrence are:

1)   I do think the paper could be made substantially shorter, with a tighter focus on the data itself and what is new (as opposed to review – while I would love to see a long review of wildfires by these authors published in another form, I think including so much background here is distracting and weakens the utility of their new material).   In addition, the way wildfire smoke interacts with PSCs or volcanic sulfate is unknown and the speculation here may be right but we really don't know. I think the material on pages 1-8, and Figures 2 and 3, would be part of a great review paper.

We discussed this point in detail and after taking into account all the questions we get during numerous oral presentations in the last 6 months, we feel that  the introductory figures 2 and 3 are simply a NEED to understand the full story. Especially if the audience of the ACP journal coming from multidisciplinary field.

We need to present the potential pathways on how smoke can be involved in all the processes that finally lead to ozone depletion. Since this is still one of the first papers on this new topic (impact of smoke on stratospheric ozone depletion), we need section 2. Currently we are still too far in thinking about review papers.

We do not see how we can skip the initial part (pages 4-8 and figures 2-3) and, at the same time, still provide a consistent and convincing  explanation of our findings.

We did largely revised and reduced the text in Sect. 2.

2)   Line 294.  Please specify what dataset(s) was used for this averaging; several were mentioned.

We are now more specific: MOSAiC data (Ohneiser2021) and DACAPO PESO data (Ohneiser2022a)

3)   Figure 4 is useful, but since a key focus of the paper is the changes at altitudes too low for normal PSC chemistry, I think it would be very valuable to add two more panels showing the same thing for the ozone in the 9-12 km region that is highlighted in the abstract and elsewhere. We really need to see how unusual the reported changes in that height range are, compared to

other years. This should be presented either here or elsewhere. Also, can 2021 be added to this plot?

Done. We added panels for the 10-12 km layer, and we included 2021.

4) Line 364.  Near-complete reduction at what vertical level?

At 18 km height. It was noted in line 365 of the original version:

5) Line 371.   Reference is needed for the statement that Raikoke contributed 10%, or say "(see below)".

Done. Ohneiser et al. (2021) is now given. We added a new section (Sect.6.2) to explain again why we think the Raikoke aerosol fraction was of the order of 10-20%.o

6) Can Figure 6 be hatched to show values that are the lowest in the record (if there are any)? Also, how can we relate this figure to statements made elsewhere regarding the 9-12 km layer?

We slightly improved Fig 6. by showing the 10-12 km layer (we switched from 9-12 km to 10-12 km in the revised version) and the PSC regime (now in the revised version from 14-23 km height). The message of the figure is to see the two ozone holes in 2020 and 2021 (very blue colors), and that the PSC evolution and ozone depletion occurred in smoke polluted air (by showing the base and top heights of the smoke layer). Figure 6 is an important introductory to  Figure 7 (the main figure of the Antarctic ozone study).

7) It would be great if Figure 7 could delineate the degree to which the high aerosol amounts in the lower portion of the profile are unusual. Would it be possible to show the range of previous years as a shaded region rather than just the single line for the mean?

See our further response to point 8.

8) Again re Figure 7, how about showing the range of temperature anomaly in 2020 and 2021 from reanalysis?   Single station data can sometimes be unrepresentative.

We revised and improved Fig. 7 significantly being inspired by all your fruitful suggestions (points 7,8,9).

We provide the shaded range for the background aerosol.

We show the year-by-year variability (ozone deviation and temperature deviation profiles for each individual year from 2010-2019) to better identify the natural variability.

We highlight the Calbuco year (dark gray lines, ozone and temperature anomaly profiles), and use the Calbuco impact as some kind of a (known) reference in the discussion of the importance of the smoke impact.

We use ERA5 temperature fields (70-90N and 70-90S) to obtain more representative temperature profiles.

The discussion is expanded accordingly.

9) Same issue for the low altitude ozone anomaly in Figure 7 – how unusual is it?   A time series

like what I suggested in Figure 4 would best answer that question in my view; alternatively, a range of variability in other years could be shown here.

See our response to point 8.

Furthermore, we used the Hofmann et al. (Nature, 1987) approach and show times of the ozone mixing ratio for Antarctica (Neumayer+South Pole) from mid-August to mid-October 2020 and the same for 2021, for the height layers from 10-12 km, 14-16 km, and 18-20 km, to see the potential impact of ozone transport. The smooth and monotonic decrease indicates the dominance of heterogeneous chemical processes. This is now presented in the new Fig.8.

10) Lines 460-463. This is a very important point – can it be expanded and substantiated?

The text in lines 460-463 was: In the lower height range (9-12 km height, below the main PSC layer), an expected clear correlation between the smoke SA concentration and an additional ozone loss was, however, not found in 2020. The reason may be related to specific meridional ozone transport and tropospheric-stratospheric exchange processes in 2020. A clear relationship between smoke occurrence and negative ozone anomaly at PSC-free conditions (9-12 km) was found in 2021, only (Fig. 7d-f).

We rearranged the text, but we still do not have a better answer (now for the 10-12 km height layer). There was certainly a smoke-related ozone loss by chemical processes (in this region the aerosol perturbation showed a maximum), but obviously this reduction was compensated by ozone transport in 2020. This is now a bit better discussed based on the new Fig.8.

11) The discussion of how much extra ozone loss occurs in the PSC region (lines 479-484) is pretty rough. Factors such as temperature and transport can induce variability in this region from year to year. This discussion is not very convincing as a result. Can you put this figure in context relative to variability from other factors? If not, I suggest deleting it.

The text in the lines 479-484 was: To evaluate the importance of an apparent smoke-related ozone reduction of 0.4-1.2 mPa (9-12 km) and 1-1.5 mPa (15-20 km) the following numbers are useful. Under volcanic widely quiescent stratospheric aerosol conditions (2010-2019) the ozone partial pressure in the central 15-20 km PSC height range dropped from values of 11.5-14.5 mPa (May-June 2010-2019 mean, Neumayer + South Pole stations) to values around 4 mPa (September-October 2010-2019 mean), and thus by 7.5-10.5 mPa (i.e., on average by 65-72%). According to this, a smoke-related further ozone loss of 1-1.5 mPa as found in September-October 2020 corresponds to a relative additional ozone reduction by 10-20%.

We deleted this paragraph.

12) Line 505. Please clarify how you identify the PSCs only about 18 km. How can you be sure?

This was just a comment to the shown lidar observations. We changed this misleading sentence.

We provide much more information on PSCs now. PSCs can be appear even at heights close to the tropopause over Antarctica (Pitts et a., 2018, Tritscher et al., 2021). Over the Arctic the PSCs were exclusively found above 14 km height (Deland et al., 2020).

13) Line 521. I see your broad points here but I don't follow how you get the fraction volcanic –

yes, the ratio would likely have been about 45 for volcanic but was measured to be 70. But please explain how you get 10%.

We introduce a new subsection on the potential Raikoke impact (Sect. 6.2). In that subsection, we present an alternative way regarding the estimation of the Raikoke fraction. This alternative estimation is in good agreement with the more sophisticated one presented by Ohneiser et al. (2021).

14) Figure 10.   Same comment as above re. the corresponding Antarctic plot, Fig 4.

10-12 km height layer is now highlighted by horizontal lines.

15) Line 543.   Same problem with variability as made above re the Antarctic; I don't think you can reliably infer the extra ozone loss in this way.   For example, cold years also imply very little downwelling, which will also lead to low ozone values, and I think you can't be confident from such a simple way of looking at it.

We rephrased and shortened the section. We avoid such quantitative statements.

16) The low altitude anomalies are key in Figure 11.   Same comments as above for the Antarctic – we need to better understand how unusual these are.

We present Fig. 11 (now Fig. 12) in the same way as the improved Fig. 7 (showing year-by-year ozone and temperature deviation profiles, and using ERA5 temp data).

17) In closing, I want to make one more major comment. There is evidence from OMPS data and presentation on that which I have seen that the aerosols from the Soufriere eruption in 2021 may have contributed to the aerosol load in the Antarctic in that year. It would on the other hand, be surprising if so much of the 2020 Australian fire aerosol lasted into 2021.  You may want to make the same kind of comparison that you did for the Arctic in Figure 9 to probe the extinction and what that can tell you. Or you may want to at least discuss the possibility.

Thank you. We introduce a new subsection (Sect. 5.3) on the potential impact of La Soufriere volcanic sulfate aerosol. However, our Punta Arena observations (Ohneiser et al., 2022a) do not support the hypothesis that La Soufriere volcanic aerosol was responsible for the ozone hole in 2021.

---

## Author Comment (AC2)

Dear Mike,

We thank you for your very careful reading and fruitful suggestions.

Please find below are our responses highlighted in BLUE.

In our revised manuscript, text that was significantly changed or new text parts that were added are given in BOLD.

Ansmann et al. (2022) (hereafter A22) focus equally on Antarctic and Arctic ozone depletion and its proffered connection to stratospheric wildfire smoke. This comment offers questions/concerns with both aspects.

Regarding the Antarctic aspect of A22, the only smoke-particle information presented is from Punta Arenas. At 53°S, this location is generally on the extratropical side of the Antarctic polar vortex. Smoke observations at such a position with respect to the polar vortex interior provide no information on aerosol conditions inside the vortex edge, an effective mixing barrier according to volumes of previous research. To make a plausible link between Punta Arenas stratospheric smoke and PSCs, the authors may need to demonstrate evidence of smoke inside the vortex, especially in the days leading up to PSC formation (typically late May). Any observation at Punta Arenas that is outside the vortex edge has no direct bearing on PSCs and PSC-related ozone depletion. It would be very interesting and important to see Punta Arenas lidar profiles segregated by their location with respect to the vortex edge. If inside-vortex aerosol enhancements consistent with the pyroCb smoke signature can be demonstrated, in air too warm for PSC support, that would immeasurably advance A22's argument.

Thank you for this comment. In Sect 5.2, we explain that we assume that the Punta Arenas time series represent well the aerosol conditions over the entire southern part of the Southern Hemisphere. We support our assumption by satellite observations presented by Rieger et al. (2021) and Yook et al. (2022). Furthermore, simulations of the spread of smoke and volcanic aerosols in the Southern Hemisphere and Northern Hemisphere corroborate the fast transport towards the polar regions within a few months (Haywood et al. 2010, Kloss et al., 2021, Zhu et al., 2018).

Regarding the potential differences between the aerosol conditions outside (Punta Arenas) and inside the vortex we can only speculate. Nobody knows! Is there an accumulation of particles (because an exchange with the air outside the vortex is not possible) or an enhanced decrease (by stronger downward transport within the vortex)? However, we use our Arctic (MOSAiC) observations to characterize the potential impact of the polar vortex on aerosol properties (accumulation of aerosol with time during the vortex lifetime, Ohneiser et al., 2021). We state that in Sect. 5.4.

Regarding the Arctic aspect of A22, they build on conclusions drawn by Ohneiser et al. (2021) (O21), namely that smoke polluted the Arctic stratosphere in 2019 and 2020. O21 hypothesized that smoke arrived in the stratosphere by a non-pyroCb pathway; Siberian smoke was diabatically lofted from the lower troposphere across the tropopause and continued its diabatic ascent thereafter. O21's confident determination of smoke altitude extent, based on the Polly lidar-ratio calculation, was capped at 12-13 km. A22 loosen that constrain in section 4.3.1: "Before we can deepen this discussion, we need to clarify that wildfire smoke was the dominating aerosol component throughout the entire stratospheric aerosol layer up to 18 or even 20 km height. Ohneiser et al. (2021) left the question open whether smoke or sulfate aerosol originating from the Raikoke volcanic eruption (Kloss et al., 2021) was prevailing at heights >13 km. Because of too noisy Polly lidar signals, wildfire smoke could unambiguously be identified up to 13 km height only." A22's presentation of Figures 8 an 9 appears to be an argument toward a reinterpretation of O21's smoke-altitude cap. The argument is based largely on an admittedly noisy HSRL AOD profile and a set of generated AOD curves spanning a selection of lidar ratios. Figure 9b reveals that a lidar-ratio selection of 60 would give a solution with significant HSRL-data overlap throughout the displayed z range. Prata et al., (2017)

demonstrated a 532 nm lidar-ratio central tendency between 59-66 sr in an analysis of two distinct stratospheric volcanic sulfate plumes (Kasatochi and Sarychev Peak). Considering the lidar-ratio error bars in Figure 9a, Prata et al. (2017), and Mattis et al. (2010), it appears that sulfates could explain the HSRL AOD profile in Figure 9b as convincingly as smoke. Given the undisputed evidence that Raikoke sulfates were a hemispheric, high northern latitude presence in 2019 and 2020 (Kloss et al., (2021); Gorkavyi et al. (2021); Cameron et al. (2021)), shouldn't equal weight be given to their establishment in the Arctic during MOSAiC?

Following your comment, to better emphasize the Raikoke aspect, we introduce a new subsection (Sec. 6.2). In this new section, we present another approach to estimate the Raikoke fraction. This approach is based on satellite observations (personal communication, Linlu Mei, Bremen University). Linlu Mei presented aerosol optical thickness (AOT) retrievals (for 60-90N) during a MOSAiC workshop (in April 2022), and showed that the 550nm AOT was close to 0.3 over the High Arctic in August 2019, around 0.15 in September 2019, 0.1 in October, and 0.06-0.07 in the beginning November 2019. These estimations are in a good agreement with our MOSAiC Polarstern observations since the beginning of October 2019.

These satellite observations together with our observations (discussed in the Ohneiser-2021 paper, lidar observations at Ny Alesund, Polarstern observations later on) point to a high UTLS AOT of 0.15-0.2 in the High Arctic in August 2019. This is described in Sect.6.2

We again discuss in Sect.6.2, as in previous publications, that according to the emitted SO2 mass of 1.5-2 Tg, the Raikoke volcanic eruption produced, to our best knowledge, 550nm AOTs of 0.025-0.03 (on a hemispheric scale). This is a contribution of 10-20% to the observed AOT of 0.15-0.2. These values are in good agreement with the more advanced approach presented by Ohneiser et al., 2021 (10-15%) (based on multiwavelength Raman polarization lidar observations).

Now to Prata et al (2017) and the CALIOP data analysis. Two things should be mentioned here. Prata et al were forced to apply the forward mode of the Klett method. This data analysis method is quite uncertain. Furthermore, they had to assume pure Rayleigh backscatter below an isolated sulfate layer. To our opinion, there are always sulfate particles below a volcanic sulfate layer. If there is sulfate backscatter below the volcanic layer, the assumption of pure Rayleigh backscattering above and below a volcanic layer leads to an overestimation of the layer-mean lidar ratios. In the case of long signal averaging (curtain profiles, much improved signal-to-noise ratio), the lidar ratio was close to 50sr (in the Prata paper) as expected for aged volcanic sulfate aerosol and would have been probably close to 40sr when keeping smoke below the layer into account.

There is no doubt for us that lidar ratios of 65-75 sr at 532nm, and 40-50sr at 355nm, clearly indicate smoke and not volcanic aerosol. These results were comprensively explained and discussed in our previous pulications (Haarig et al., 2018, Ohneiser et al., 2020, 2021, 2022).

---

## Author Comment (AC3)

Dear Reviewer!

We thank You for careful reading and some fruitful suggestions.

Our answers are in BLUE!

Significantly changed text or added text is given in BOLD in the revised version of the manuscript.

The paper aims to highlight and quantify the role of extreme events of wildfire smoke to the ozone depletion in the Arctic and Antarctic. The study is based on lidar measurements concerning the evolution of the smoke in the stratosphere and on ozonesondes concerning the vertical distribution of ozone. The authors further analyze a unique lidar dataset, already presented in previous studies, concerning the presence of wildfire smoke in the polar stratosphere and aim to associate this with ozone depletion in the Arctic and Antarctic.  They provide a very comprehensive review on the impact of various types of stratospheric aerosols on PSC formation and attempt to describe the potential impact of aged smoke on PSC formation. They also provide a very detailed analysis of the characteristics of the aerosol layers and the ozone departures at various layers from ozonesonde measurements. The paper is very interesting, well written and structured and it is of high importance, concerning the potential impact of possible increasing extreme wildfire events on ozone loss. The paper should be accepted for publication but some open questions/issues should be addressed or considered in a revised version.

The authors present vertically and temporally resolved ozone anomalies using the 2010-2019 period as a reference.  Is the choice of this period arbitrary? They don't provide any significance of these anomalies and it is hard to judge to what extent the event they present are unique and unexpected.

We discuss the selection of the 2010-2019 time period as reference period in Sect. 3.2 and 4. The period is selected because trend effects are visible in the Antarctic ozone time series. Also other groups concentrate on periods such as from 2012-2019 (Rieger et al., 2020, Yook et al., 2020). So we decided to use 2010-2019 for both, Arctic and Antarctica.

The authors associate ozone anomalies with ozone loss but they don't mention what is the contribution of dynamics in the observed anomalies. I believe that the authors claim that there is an excess chemical loss due to the presence of wildfire smoke compared to previous years. How do they exclude the impact of different dynamics. How significant are such changes (in dynamics) compared to the climatology during the years under study? These is not clear in the analysis.

We assume that transport effects and trend effects are widely eliminated when averaging ten years of data… as written in Sect. 4. Then, any ozone deviation from the mean ozone profile should indicate the smoke impact. We improved the key figures (Figures 7 and 12) as suggested by Susan Solomon so that one can see the year-to-year springtime ozone variability, and the impact of temperature (on PSC volume) and transport (dynamics). And for Antarctica, the dynamics impact is likewise low, and very large for the Arctic.

The authors show relative occurrence of PSCs from CALIPSO. How does this compare to average values? Is this frequency larger than the previous years? And if yes how is this associated solely to the presence of smoke?  Please comment on this.

We mention in Sect. 5.4 that the PSC amount we derived from the CALIOP data varied not much from 2015 -2021, within just 10% around the mean.